

**Subtropical gyre persistence in the Gulf of Cadiz, southern Iberian margin, interrupted by extremely cold surface water incursions during the Early – Middle Pleistocene Transition**

Aline Mega[1,2] (0000-0002-9386-2261), Teresa Rodrigues[1,2] (0000-0001-7811-7506), Emília Salgueiro[1,2] (0000-0003-1000-2977), Mária Padilha[1] (0000-0002-7103-5695), Henning Kuhnert[3] (0000-0001-5242-4495), Antje H. L. Voelker[1,2] (0000-0001-6465-6023)

[1]Divisão de Geologia e Georecursos Marinhos, Instituto Português do Mar e a Atmosfera (IPMA), Avenida Doutor Alfredo Magalhães Ramalho 6, 1495-165 Alges, Portugal.

[2]CCMAR Associated Laboratory, University of the Algarve, Campus de Gambelas, 8005-139 Faro, Portugal.

[3]MARUM, Universität Bremen, Leobener Straße 8, 28359 Bremen, Germany.

*Correspondence to*: Aline Mega (alinemega20@gmail.com)

**Abstract.** Besides the shift in dominant orbital cyclicity, the mid-Pleistocene Transition or Early-Middle Pleistocene Transition (EMPT) was characterized by a change in the deep thermohaline circulation. Those changes contributed to more intense and longer-lasting glacial periods and cooler sea surface temperatures (SSTs). Within the Atlantic Ocean, the Iberian margin is considered a key location to study climatic variations influenced by both high- and low-latitude processes. In this study we focus on IODP Site U1387 on the southern Portuguese margin to reconstruct surface water circulation and related plankton foraminifera ecosystem changes during the interval of Marine Isotope Stage (MIS) 28 to MIS 18 (1006-750 ka). Our planktonic foraminifera assemblages and SST reconstructions (foraminifera assemblages and $U^{K'}_{37}$ alkenone index) demonstrate warm, stable SST conditions during much of the interval due to persistent influence of subtropical gyre waters as indicated by the tropical-subtropical and Azores Current related foraminifera species and the periods with dominant sinistral coiling direction of the species *Globorotalia truncatulinoides*. Maximum interglacial SSTs were up to 2ºC warmer than at present in both summer and winter, with the exception of interglacial MIS 23 with SSTs ~1.5°C colder than in the other interglacials. Subsequent the respective glacial inception, the relative warm conditions were periodically interrupted by millennial-scale extreme cold events when polar species *Neogloboquadrina pachyderma* became abundant



(>30%) and the SSTs, reconstructed from the foraminifera assemblage data, dropped below
10°C in summer and 5 °C in winter. The most pronounced event, considering the amplitude of
cooling and duration, occurred between 870 to 864 ka, marking the terminal stadial event of
the MIS 22/MIS 21 transition (Termination X). Extreme cold events, always associated with
the incursion of subpolar waters into the Gulf of Cadiz, mark all the terminal stadial events
from Terminations XII to IX and the millennial-scale variability during the transitions to full
glacial conditions, although the duration of the cooling varied greatly. The extreme cooling
was only possible through migration of the subarctic front into the lower mid-latitudes as a
consequence of an extreme reduction in the Atlantic meridional overturning circulation. The
amplitude of cooling, duration, and frequency of subpolar water incursions during MIS 24 to
MIS 22 stands out, providing further evidence for the "900 ka event" being a key feature of the
EMPT.

**1. Introduction**
A major global climatic shift, known as the mid-Pleistocene or Early-Middle
Pleistocene Transition (EMPT), took place between 1250 and 650 thousand years (ka) ago,
dramatically changing Earth's climate dynamics (Clark, 2012; Clark et al., 2006; Head and
Gibbard, 2015; McClymont et al., 2013). This period was characterized by long-term cooling
in global mean sea surface temperatures (SSTs), lower glacial atmospheric carbon dioxide
levels and a change in the deep-water circulation, stratification and carbon storage during the
glacial periods that ultimately resulted in more intense and longer-lasting glacial periods
(changing from 41 kyr to 100 kyr cycles) and cooler SSTs (Chalk et al., 2017; Clark et al.,
2024; Farmer et al., 2019; Kim et al., 2021; Tachikawa et al., 2021; Willeit et al., 2019;
McClymont et al., 2013). The major shift in the deep-water circulation during the EMPT, often
considered as the first 100 ka cycle, is referred to as "the 900 ka event" (Farmer et al., 2019;
McClymont et al., 2013; Pena and Goldstein, 2014).
The causes of these long-term patterns of Quaternary climate have been attributed to
internal changes in climate response to orbital forcing, as the latter did not change over this
time (Clark, 2012; Clark et al., 2006; Hodell and Channell, 2016; Shacketon, 2000). It is
believed that the EMPT may have been influenced by ocean-atmosphere system changes, with
declining atmospheric carbon dioxide concentrations and continental ice-sheet growth playing
a role (Chalk et al., 2017; Willeit et al., 2019). During the EMPT glacials, lower sea-levels
contribute to benthic $\delta^{13}$C values reaching their lowest levels in 5 million years (Westerhold et
al., 2020), which may be caused by exposed continental shelves accelerating the transport of



organic carbon into the oceans (Head and Gibbard, 2015). Nowadays, water masses carrying
lower $\delta^{13}C$ signals (<0.5 ‰) are formed by convection around Antarctica (Antarctic
Intermediate Water, Antarctic Bottom Water/AABW) and spread out into the global ocean
basins (Curry and Oppo, 2005; Kroopnick, 1985). Northward and upward expansion of such
signals in the Atlantic basin during glacial periods was therefore interpreted to reflect the
replacement of North Atlantic Deep Water (NADW) by southern sourced waters and thus a
reduced Atlantic Meridional Overturning Circulation (AMOC) (Hodell and Channell, 2016;
Raymo et al., 2004; Raymo et al., 1990; Sarnthein et al., 1994). Weakening NADW influence
throughout the MIS 22-MIS 24 interval is supported by neodymium isotope records (Farmer
et al., 2019; Kim et al., 2021; Pena and Goldstein, 2014; Tachikawa et al., 2021). A possible
explanation for the increase in glacial $\delta^{18}O$ values during the "900 ka event" relates a weak
AMOC and low insolation in the Southern Hemisphere during Marine Isotope Stage (MIS) 23
to maximum continental ice volume build-up, which continued to be registered in the
subsequent glacials (Elderfield et al., 2012; Pena and Goldstein, 2014).
Most of the water stored during Quaternary glaciations in the Laurentide, Greenland
and European ice sheets was discharged into the North Atlantic Ocean during the last 1.5 Ma,
producing short cold events that were often associated with ice-rafted debris (IRD) deposition
(Barker et al., 2022; Barker et al., 2021; Hodell and Channell, 2016; Jansen et al., 2000). The
effect of ice-cover changes during the EMPT, mainly associated with the "900 ka event", has
been reported based on different proxies and in the (sub)polar regions of both hemispheres. In
the North Atlantic, Wright and Flower (2002) found extremely cold events from 1000 to 500
ka at ODP Sites 980 (55ºN, 15°W) and 984 (61ºN, 24°W) (Fig. 1), based on the percentage of
polar species *Neogloboquadrina pachyderma* and IRD records, later on corroborated by the
1.7 Ma long records for Site 983 (60°N 24°W) (Barker et al., 2011; Barker et al., 2022). These
data, in conjunction with increased reworked nannofossil abundance during the IRD events at
Sites 980/981 (Marino et al., 2011), suggest that the Arctic front shifted from a position
between those Sites southward and the sea-ice cover expanded greatly during those periods as
a result of reduced NADW production. That scenario is supported by evidence from IODP Site
U1314 (56.36°N, 27.88°W), where Hernández-Almeida et al. (2013) observed an abundance
of *N. pachyderma* of up to 93 % during the "900 ka event". Between 900 and 675 ka, the same,
short-term extreme cold events were registered further south at IODP Site U1385 (Iberian
Margin) as cold SST events associated with lower salinities (higher percentages of the C37:4
alkenone) (Rodrigues et al., 2017). All those cold events were associated with a northward and
upward penetration of AABW and thus reduction in the AMOC depth (Hodell and Channell,



2016; Hodell et al., 2023a; Hernández-Almeida et al., 2015), especially during the terminal
stadial events.

The western Iberian margin is a key area for high-resolution paleoclimatic studies
because it is climatologically sensitive to high and low latitude processes. Following the
seminal work of Shackleton et al. (2000), it is known that benthic foraminifera $\delta^{18}O$ records
from depths greater than 2500 m on the southwestern Portuguese margin reflect an Antarctic
climate signal, in particular Antarctic temperature variations, whereas surface water records
from the western and southern Portuguese margin mimic the millennial-scale Greenland
interstadial/stadial oscillation and thus record northern hemisphere temperature variations. This
concept has now been proven for the last 1440 ka with the high-resolution records of IODP
Site U1385 (Hodell et al., 2023a).

Furthermore, planktonic foraminifera assemblages are reliable sources for
environmental conditions in the western Iberian margin and specific assemblages can identify
prevailing oceanographic conditions. At modern conditions, subtropical species, among them
*Globigerinoides ruber* white, reflect the influence of the Azores Current (AzC), whereas
*Globigerina inflata* and *Neogloboquadrina incompta* represent the Portugal Current and
*Globigerinoides bulloides* upwelling events (Salgueiro et al., 2008). Increased abundances of
*Turborotalita quinqueloba* and *Neogloboquadrina pachyderma*, on the other hand, can provide
insights into past incursions of subpolar waters and southward displacement of the subarctic
front (boundary between the subtropical and subpolar gyres) (Eynaud et al., 2009; Girone et
al., 2023; Johannessen et al., 1994; Martin-Garcia et al., 2015; Pflaumann et al., 2003;
Salgueiro et al., 2010; Singh et al., 2023).

Recent studies (Bajo et al., 2020a; Voelker et al., 2015) confirmed extremely cold SST
conditions during stadial climate events of the EMPT also at southern Portuguese margin IODP
Site U1387 (Fig. 1). However, detailed information on the surface-water conditions during the
"900 ka event" (MIS 24 to MIS 22) and during the lead up to it remains limited. This study,
therefore, aims to characterize surface-water conditions at IODP Site U1387 between MIS 28
and MIS 18 (1006-750 ka) to better understand the climate dynamics and oceanographic
changes that occurred during this critical period. Situated in the northern Gulf of Cadiz, Site
U1387 is highly sensitive to changes in the North Atlantic subtropical gyre and to the water
mass exchange between the North Atlantic and the Mediterranean Sea. Moreover, the high
sedimentation rates ($\geq 20$ cm kyr$^{-1}$) in contourite drifts like the Faro drift, into which Site U1387
was drilled, provide exceptional paleoclimate records with high temporal resolution
(Hernández-Molina et al., 2016b). For evaluating temperature changes, both in terms of



amplitude and timing, and their relationship to the prevailing oceanographic conditions, we
produced high-resolution, sub-millennial-scale records of planktonic foraminifera assemblages
and SST reconstructions. Using a multi-proxy approach, the SSTs were reconstructed in two
ways: 1) converting the planktonic foraminifera assemblages into summer and winter SSTs
using a transfer function; and 2) based on the $U^{K'}_{37}$ alkenone index, approximately reflecting
annual mean SSTs. Strength in subtropical gyre circulation was inferred from the dominant
coiling direction of the planktonic foraminifera species *Globorotalia truncatulinoides* (Billups
et al., 2016). We compare our data with other available records from the southwestern Iberian
margin, as well as sites from the mid-latitudinal North Atlantic. This comparison allows us to
contextualize our results within broader regional and global climatic trends, providing insights
into the variability and connections between these key areas during the study period. By
integrating these records, we aim to improve our understanding of both local and large-scale
processes affecting this Northeast Atlantic region.

**2. Regional Setting**

The subtropical gyre nowadays comprises much of the surface and sub-surface waters

in the low-to mid-latitudinal North Atlantic, is approximately 1000 km in diameter and
distributes heat and moisture to the north (Fig. 1A). The gyre circulation is driven by a
combination of trade winds, westerlies and the Coriolis force, whereby the westerlies dominate
the circulation in its northern part, especially during the winter. The strength and position of
the oceanic currents depend, therefore, on the variability of the atmospheric wind fields. During
the winter the latter are characterized by the eastward displacement of cyclonic perturbations
(Relvas et al., 2007).

Located within the southern mid-latitudinal North Atlantic, the Gulf of Cadiz has a

surface-subsurface current system dominated by three branches of the North Atlantic's
subtropical gyre circulation: the eastward flowing AzC between 34.3 and 35.7ºN, contributing
with heat and salt; the Azores Counter-Current between 37.74 and 39.24ºN and the Canary
Current that flows south-westwards (Carracedo Segade et al., 2015). The AzC dominates the
Gulf of Cadiz surface waters (0 to 500 m) and partially recirculates along the western Iberian
margin through the Iberian Poleward Current that results from the seasonal reversal of the wind
regimes (Frouin et al., 1990; Peliz et al., 2005) (Fig. 1B). Also, the Gulf of Cadiz is an important
transition zone where the Mediterranean Outflow Water flows at intermediate depth level,
adding high salinity and heat to the North Atlantic Circulation (Ambar et al., 1999; Folkard et
al., 1997).



The Gulf of Cadiz receives contributions from the Portugal Current and the Portugal
Coastal Current (Fiuza et al., 1998). The Portugal Current flows equatorward transporting
cooler and less saline waters into the region (Carracedo et al., 2014; Peliz et al., 2009). The
Portugal Coastal Current exists only during the upwelling season from late May/early June to
late September/early October, driven by the northerly winds that transport cold and less saline
upwelled water (jet-like) southward (Criado-Aldeanueva et al., 2006; Folkard et al., 1997) (Fig.
1B). Near Cape São Vicente, a part of the Portugal Coastal Current jets turns eastward under
favorable wind conditions and enters the Gulf of Cadiz flowing along the upper slope toward
the Strait of Gibraltar interacting with the upwelling off Cape Santa Maria (Sanchez and
Relvas, 2003) and affecting the region of Site U1387.
The Gulf of Cadiz SSTs have a seasonal behavior observed by Folkard et al. (1997)
through satellite images. Temperatures vary between 22.5 °C (summer) (Fig. 1B) and 16.5 °C
(winter) with a mean value of 19.6 °C (Vargas et al., 2003).

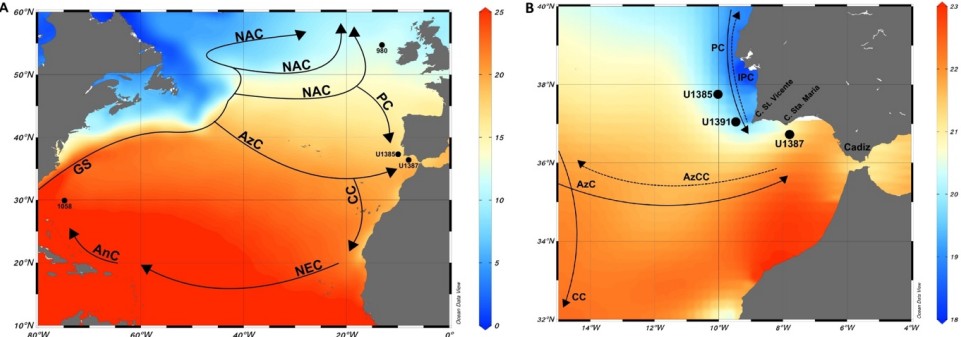


**Figure 1:** A: North Atlantic Ocean with annual mean SSTs (°C) at 0.25-degree resolution as
background (WOA 2023; Reagan et al., 2024). Location of IODP Site U1387 and other
available North Atlantic records discussed in the text (IODP Site U1385; ODP Site 980; ODP
Site 1058; DSDP Site 607/IODP Site U1313). Black arrows represent the surface circulation:
GS – Gulf Stream; NAC – North Atlantic Current; PC – Portugal Current; CC – Canary
Current; AzC – Azores Current; NEC – North Equatorial Current; AnC – Antilles Current.
B: Close-up of the study area with locations of IODP Site U1387 and SW Iberian Margin IODP
Sites U1385 and U1391 with the mean summer (July-September) SSTs (°C) at 0.25-degree
resolution as background (WOA 2023; Reagan et al., 2024). Black arrows represent the surface
circulation: AzCC – Azores Countercurrent; IPC – Iberian Poleward Current. Currents adapted
from Baptista et al. (2021) and references therein. Background maps made with ODV
(Schlitzer, 2023).



## 3. Material and Methods

IODP Site U1387 (36°48.3210'N, 7°43.1321'W) was drilled in December 2011 by the Integrated Ocean Drilling Program (IODP) during Expedition 339 - Mediterranean Outflow into the Faro Drift, northern Gulf of Cadiz, at a water depth of 559 m (Fig. 1) (Expedition 339 Scientists, 2013). The samples were collected at a resolution of 12-13 cm along the revised splice (Voelker et al., 2018), except for the interval of Termination X where the resolution was increased to 6-7 cm for the Bajo et al. (2020a) study. Each sample was freeze-dried, weighted and washed through a 63 µm-mesh sieve, following the procedure established in the Sedimentology and Micropaleontology Laboratory of the Division for Geology and Marine Georesources at the Portuguese Institute for the Sea and Atmosphere (IPMA) (Voelker et al., 2015). The coarse fraction residue was transferred onto filter paper, dried at 40 °C, and weighted.

### 3.1 Stable isotope measurements

To establish a stable oxygen isotope record for the chronostratigraphy, 6-12 specimens of the planktonic foraminifera *Globigerinoides bulloides* were collected from the fraction >250 µm of a total of 706 samples. The specimens were sent to the gas isotope ratio mass spectrometry laboratory at MARUM (University Bremen), Germany, where they were analyzed with a Finnigan MAT-251 or MAT-252 mass spectrometer coupled to an automated Kiel I or Kiel III carbonate preparation system, respectively. The mass spectrometers' long-term precision is ±0.07 ‰ for $\delta^{18}O$ based on repeated analyses of internal (Solnhofen limestone) and external (NBS-19) carbonate standards. Some of the isotope results were already published in Bajo et al. (2020a) and are available as Bajo et al. (2020b), although the age model used in the current study differs from those data.

### 3.2 Planktonic foraminifer assemblage analysis and SST calculations

For the planktonic foraminifera assemblage, a total of 356 samples were analyzed at a sample resolution of 24-25 cm. Each sample was dry sieved to obtain the fractions >250 µm and 150-250 µm. The respective fraction was then split until about 200 specimens remained in the fraction >250 µm and about 100 specimens in the 150-250 µm fraction. Specimens, including identifiable fragments, were counted, and identified in full in each sub-split.

Species identification followed Kučera (2007) and Schiebel and Hemleben (2017). All sinistral coiling *Neogloboquadrina pachyderma* specimens were assigned to *N. pachyderma*, in agreement with the observed morphotypes being similar to those typically found in polar



regions (supplementary figure 1). We are using the percentage of *N. pachyderma* to identify
cold water incursions of subpolar origin into the Gulf of Cadiz. The assemblage data were
converted into relative abundances (percentages) and species grouping into
tropical/subtropical, transitional, subpolar/polar habitats according to Kučera (2007). The
Azores Current factor was calculated following Salgueiro et al. (2008) and combines the
percentages of *Globorotalia inflata*, *Globigerinoides ruber* (white) and *Trilobatus sacculifer*.

To evaluate changes related to the subtropical gyre influence, we used the newly

developed proxy of the coiling direction of planktonic foraminifera *G. truncatulinoides,* which
is a subsurface dwelling species with five morphotypes. The morphotype type II is exclusive
of the Atlantic Ocean and the Mediterranean Sea and is the only type with dextral and sinistral
forms (de Vargas et al., 2001; Ujiié et al., 2010). According to Billups et al. (2016), the amount
of sinistral coiling direction of this species increases when the subtropical gyre circulation is
more intense. For this, we analyzed whenever possible all the individuals in the fraction >250
µm in all the samples where this species was found (total of 332 samples). Intervals with a high
sample volume were split before size fractioning. The coiling ratio was obtained using the
following formula: % GTS = $GTS*100*(GTS+GTD)^{-1}$ where GTS is the number of sinistral
specimens and GTD the number of dextral specimens (Billups et al., 2016; Ducassou et al.,

2018).

Using the relative abundance data in the assemblages, we estimated the SST for winter

and summer using the non-distance-weighted (ndw) option of the SIMMAX program
(Pflaumann et al., 1996), similar to the Modern Analog Technique (MAT), following Salgueiro
et al. (2014). Although Jonkers and Kučera (2019) recently showed that only 10 species
dominate the SST calculations, we used the complete set of 27 species utilized by Pflaumann
et al. (1996) to be consistent with previous reconstructions in the region (Salgueiro et al., 2014;
Salgueiro et al., 2010). SST was calculated using 10 nearest neighbors and the modern analog
database compiled by Salgueiro et al. (2014), which combines the North Atlantic database of
the MARGO project (Kučera et al., 2005) with additional samples for the Iberian Margin
(Salgueiro et al., 2008) and off NW Africa (Salgueiro et al., 2014; Voelker and Salgueiro,

2017).


**3.3 Alkenone SST reconstructions**

We also reconstructed SSTs based on the alkenone $U^{K'}_{37}$ index. Alkenones are lipid

molecules that are synthesized by coccolithophorid (phytoplankton) and can be extracted from
marine sediments using organic solvents. Lipid molecules analyses were done at 24-25 cm



resolution (same levels as planktonic foraminifera assemblage data), except for Termination X
where the resolution increased to 6-7 cm (Bajo et al., 2020c). Lipid biomarker extraction was
done in 338 samples, whereby the SST for the 216 samples between 212.3 and 257.9 c-mcd
were already published in Bajo et al. (2020a). Extraction of lipid molecules from freeze-dried
sediments followed the procedure established in the DivGM's Biogeochemistry Laboratory
(Rodrigues et al., 2017; Voelker et al., 2015), which is based on Villanueva et al. (1997). The
di-, tri- and tetra-unsaturated alkenones of 37 carbon atoms were analyzed in a Varian Gas
chromatograph Model 3800 equipped with a septum programmable injector and a flame
ionization detector (GC-FID) with a CPSIL-5 CB column. Hydrogen was used as carrier gas
at a flow rate of 2.5 ml/min and n-hexatriacontane as an internal standard to determine
concentrations. To estimate SST's, we used the $U^{K'}_{37}$ index based on the di- and tri-unsaturated
alkenones ratio and converted it into temperature values using the global core top calibration
of Müller et al. (1998), with an analytical uncertainty of ±0.5 °C.

**4. Chronostratigraphy and age models**

One goal of IODP Expedition 339 was always to use the open ocean records from Site

U1385 to establish age models for the contourite sites, which are potentially affected by current
sorting and tectonics (Hernández-Molina et al., 2016a) and are too shallow to record a global
ocean benthic $\delta^{18}O$ signal. So, for contourite sites like IODP Site U1387 the planned approach
was to correlate their *G. bulloides* $\delta^{18}O$ surface water record with the one of Site U1385 and
thus to transfer the U1385 age model(s) to the contourite site, under the assumption that those
records would be similar in such a narrow region affected by the same surface water masses.
That approach was followed in this study using the high-resolution *G. bulloides* $\delta^{18}O$ record
(Hodell et al., 2023b) published by Hodell et al. (2023a) as correlation target for the Site U1387
record. One of the age models of Site U1385 was established by tuning its benthic $\delta^{18}O$ record
(Hodell et al., 2023a) to the benthic LR04 stack (Lisiecki and Raymo, 2005), whereas an
alternative age model applied tuning to the Probstack (Ahn et al., 2017). The age model used
throughout this manuscript for Site U1387 uses the LR04 related ages, although, following
Hodell et al. (2023b), Probstack based ages (supplementary figure 2) will also be provided with
the data uploaded to the PANGAEA world data center. For MIS boundaries we follow Lisiecki
and Raymo (2005) and for MIS substage nomenclature Railsback et al. (2015).





## 5. Results

### 5.1 *G. bulloides* δ¹⁸O record and chronostratigraphy

Besides the glacial-interglacial cycles of MIS 28 to MIS 18, the *G. bulloides* δ¹⁸O record of Site U1387 reveals millennial-scale stadial-interstadial oscillations, especially following interglacial MIS 25e, MIS 21g, and MIS 19c (Fig. 2A). Notably, an interstadial event occurs within MIS 22 that is also well captured in the Corchia cave δ¹⁸O record (Bajo et al., 2020a). Overall, the record mimics the one of Site U1385 facilitating the tuning and age model transference (Fig. 2; supplementary fig. 2). The resulting age model for Site U1387 reveals that sedimentation rates were lower during the interglacial intervals dropping to values around 10 cm kyr⁻¹ (MIS 19c, MIS 21g), whereas they increased during transitional and glacial periods (Fig. 2C). The same pattern in sedimentation rates is generally observed for the Probstack based age model (supplementary figure 2), although age ranges are shifted towards younger ages in the MIS 21 to MIS 28 interval and there occurs an interval with higher sedimentation rates in early MIS 26.

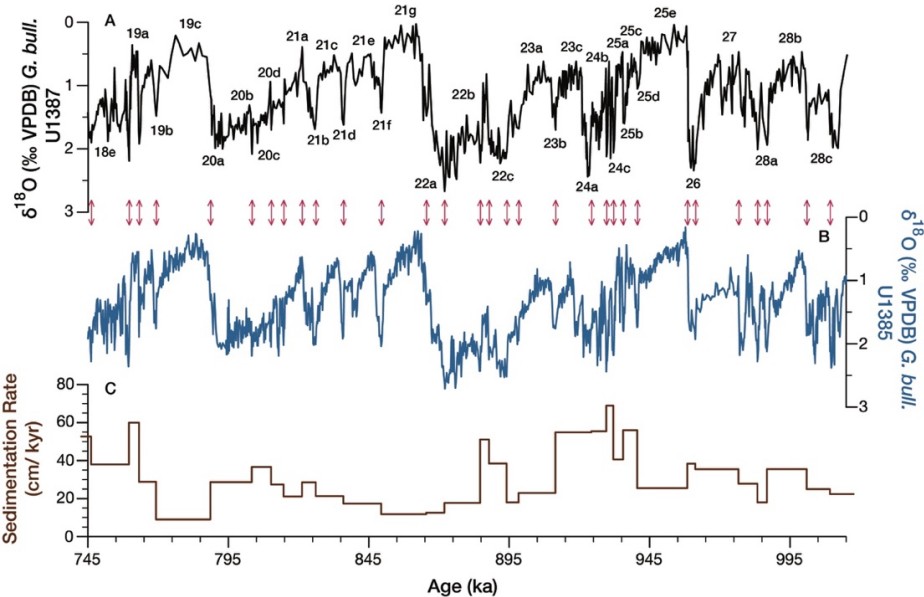

**Figure 2:** A: δ¹⁸O (‰) *G. bulloides* from IODP Site U1387 and Marine Isotopic Stages and Substages. B: δ¹⁸O (‰) *G. bulloides* from IODP Site U1385 on its LR04 related age model (Hodell et al., 2023b). Arrows between A and B indicate the tuning points between the two records. C: Sedimentation rates (cm/kyr) for IODP Site U1387.



**5.2 Planktonic foraminifera fauna**


At Site U1387, we found faunal assemblages composed by a mix of species from polar,
subpolar, transitional, subtropical, and tropical provinces (Table 1). In total, 16 species were
identified (Table 1; Fig. 3), with the diversity of the subtropical fauna appears to be diminished
due to the absence of *Globoturborotalita tenella*, *Globoturborotalita rubescens*, and
*Globorotalia hirsuta*. Although occurring in low percentages, all three species are present in
surface and Holocene aged sediments of the southwestern Portuguese margin and in the Gulf
of Cadiz (e.g., Ducassou et al., 2018; Rufino et al., 2022; Salgueiro et al., 2008), and both
*Globoturborotalita tenella* and *Globoturborotalita rubescens* have been observed in MIS 19
and younger sediments at Site U1385 (Girone et al., 2023; Martin-Garcia et al., 2015).

**Table 1** Species found at IODP Site U1387 and the respective provinces. * indicates species
associated with the Azores Current by Storz et al. (2009).

| Province | Species |
|---|---|
| Polar | *Neogloboquadrina pachyderma* |
| Subpolar | *Neogloboquadrina incompta* |
| | *Turborotalita quinqueloba** |
| Transitional | *Globorotalia inflata** |
| | *Globorotalia scitula** |
| | *Globigerinita glutinata** |
| | *Globigerina bulloides** |
| Subtropical | *Globigerinella calida* |
| | *Globigerinella siphonifera** |
| | *Globigerinoides ruber (white)** |
| | *Neogloboquadrina dutertrei* |
| | *Globorotalia truncatulinoides** |
| | *Globigerina falconensis** |
| | *Orbulina universa* |
| Tropical | *Trilobatus sacculifer** |
| | *Globorotalia crassaformis* |


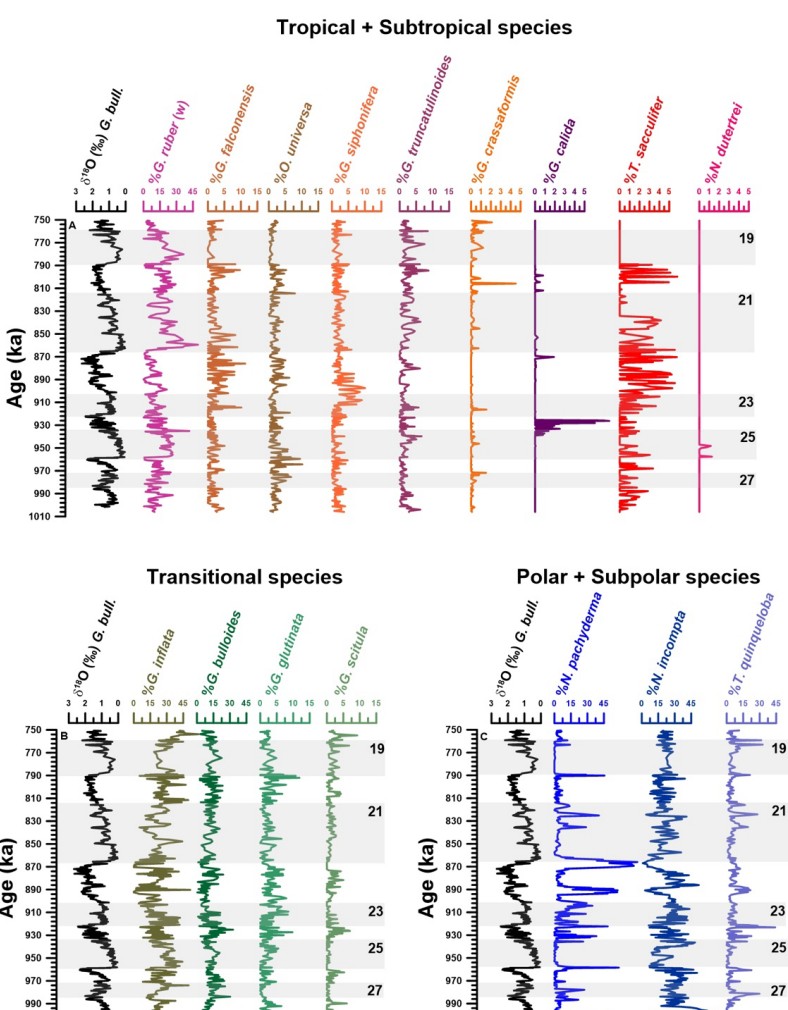

**Figure 3:** Planktonic foraminifera assemblage from IODP Site U1387. $\delta^{18}$O *G. bulloides* record (‰ VPDB) (black) provided in all three panels as stratigraphic reference. A: Abundance (%) of tropical species (*Trilobatus sacculifer*; *Globorotalia crassaformis*) and subtropical species (*Globigerinella siphonifera*; *Globigerinoides ruber* (white); *Neogloboquadrina dutertrei*; *Globigerinella calida*; *Orbulina universa*; *Globigerina falconensis*; *Globorotalia truncatulinoides*). B: Abundance (%) of transitional species (*Globorotalia inflata*; *Globorotalia scitula*; *Globigerinita glutinata*; *Globigerina bulloides*). C: Abundance (%) of polar species (*Neogloboquadrina pachyderma*) and subpolar species (*Neogloboquadrina incompta*; *Turborotalita quinqueloba*). Gray bars mark odd-numbered MIS, which include the interglacial periods. Note, differing y-axis scales.



Among all species found, only seven have average abundances greater than 2% over

the period studied, i.e. *N. pachyderma, N. incompta, G. inflata, G. ruber* (white)*, T.*
*quinqueloba, G. bulloides,* and *G. glutinata* (Fig. 3). These seven species are among the top 10
ranked by importance for transfer function models (Jonkers and Kučera, 2019). Two additional
species from the top 10 list (*T. sacculifer, N. dutertrei*) are present in the samples, but with an
abundance of less than 2 %, and one (*G. ruber* pink) is absent. In summary, the results show
an alternation of dominance between cold, transitional and warm species through MIS 28 to
MIS 18 representing changing conditions in the North Atlantic subtropical gyre.

In general, the transitional group is the more abundant one, with an average abundance

of 40.3 %, followed by the polar-subpolar group, 38.8 %, and finally the tropical-subtropical
group, 20.2 % (Fig. 4; supplementary table 1). The transitional group is present throughout the
studied interval but exhibits behavior like the tropical-subtropical group, i.e. low percentages,
during some events when the polar-subpolar group dominates the assemblage. Throughout
most of the warm periods of the record, the dextral form of the subtropical species *G.*
*truncatulinoides* dominates the coiling ratio (% GTS), with a range between 98 and 100 % (Fig.
4B). The first interval with increased contributions of the *G. truncatulinoides* sinistral form to
the total of *G. truncatulinoides* specimens occurred between 997.8 to 989.9 ka (2.4-44.4 %)
followed by seven other events: 986.8 to 981.5 ka (4-47.7 %); 966.6 to 961.6 ka (11.9-65.5
%); 958.2 to 956.9 ka (20.6-32.8 %); 930.3 to 925.1 ka (7.6-98.9 %); 887.5 to 873.3 ka (3.3-
96.9 %), followed by a short one from 871.9 to 870.3 ka (14.3-83.3 %); a double peak from
867 to 863.9 ka (40-91.7 %) and 862.7 to 855.4 ka (13.1-98.9 %); and finally 825.2 to 821.6
ka (27.9-90.8 %) (Fig. 4B). During most of those %GTS maxima, the relative abundance of *G.*
*truncatulinoides* in the assemblages increased as well (Fig. 3).




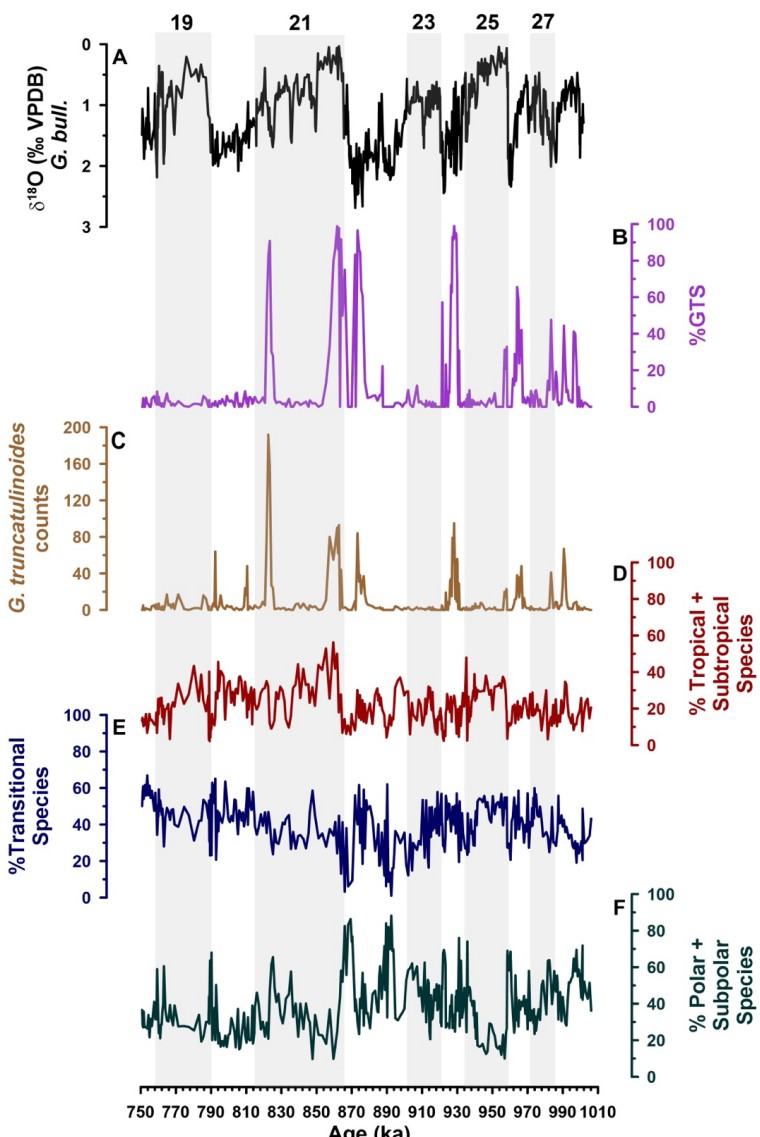

**Figure 4:** Site U1387 faunal provinces and *Globorotalia truncatulinoides* results. A: δ$^{18}$O (‰)
*G. bulloides*. B: *G. truncatulinoides* coiling ratio expressed as % of *G. truncatulinoides*
(sinistral). C: Number of *G. truncatulinoides* specimens counted in the fraction >250 µm and
used to calculate the coiling ratio in C. D: Abundance (%) summed up for Tropical and
Subtropical species. E: Abundance (%) of Transitional species. F: Abundance (%) of Polar and
Subpolar species. Grey bars mark the odd-numbered MIS.





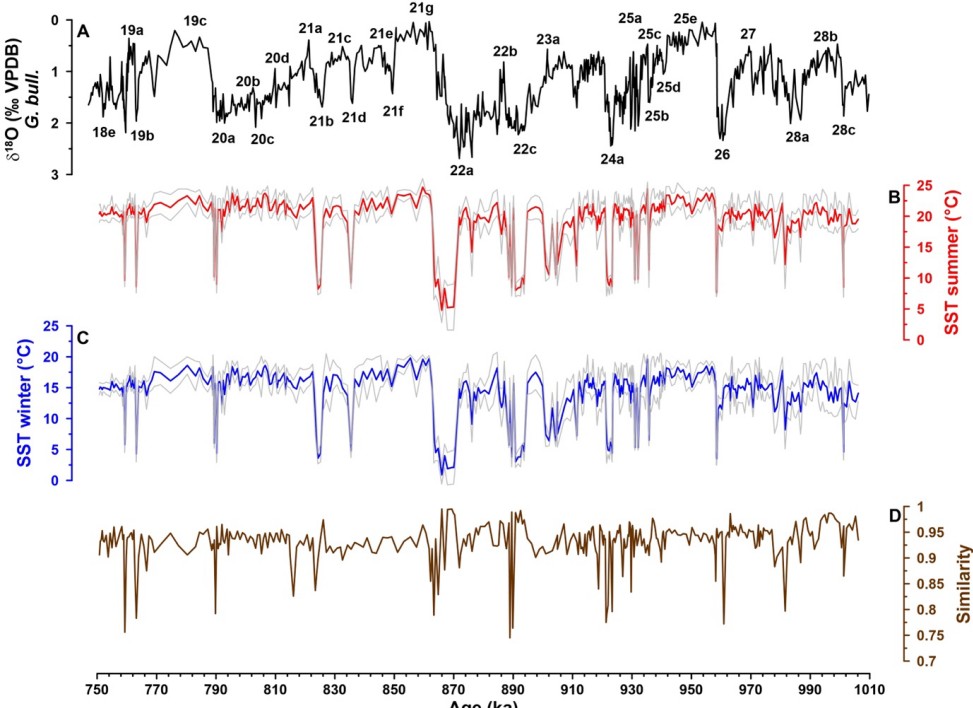

**Figure 5:** IODP Site U1387 planktonic foraminifera derived SST records A: $\delta^{18}$O *G. bulloides* (‰ VPDB) with numbered Marine Isotopic Stages and Substages. B: Summer SSTs (°C) with standard deviation (1σ). C: Winter SSTs (°C) with standard deviation (1σ). D: Similarity to modern analogs used to calculate the SSTs.

**5.3 Sea-Surface Temperatures**

The SSTs from IODP Site U1387 estimated with the planktonic foraminifera assemblage (PF-SST) do not reflect a clear pattern for interglacial-glacial cycles from MIS 28 to MIS 18. Although winter SSTs varied between 0.9 to 19.8 °C and summer SSTs between 4.8 to 24.7 °C (Fig. 5), temperatures remained elevated and relative stable during long periods with an average of 14.3 °C for winter and of 19.4 °C for summer, excluding the extreme cold events. Those conditions were interrupted by extreme short cold events when the percentage of polar and subpolar species increased (Fig. 4F), and winter SSTs dropped below 5 °C (Fig. 5C). Such events occurred within the following MIS substages: 28c; 28a; 26; 25b; 25a; 24c; 24a; 23b; 23a; 22c; 22a; 21d; 21b; 20a; and 19b. The error (1σ) for the winter SSTs ranges from 0.2 to 4.8 °C and for the summer SSTs from 0.3 to 5.4 °C, with the larger errors associated

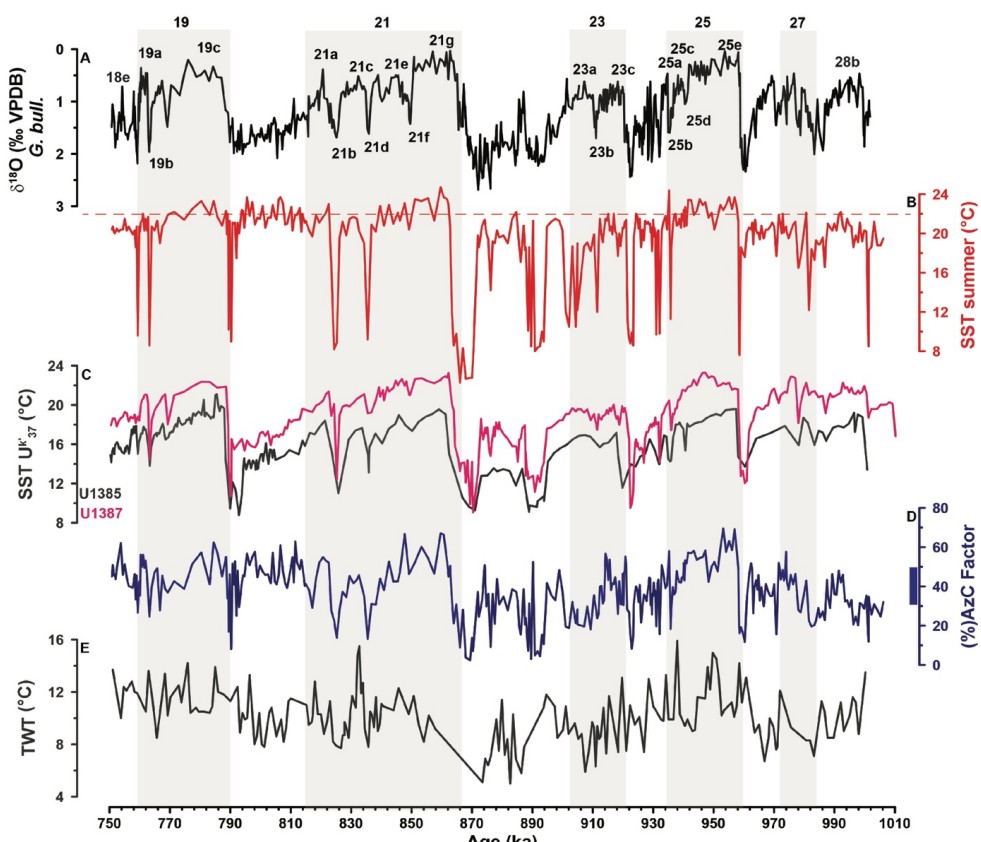

**Figure 6:** Comparing Site U1387 and Site U1385 temperature records. A: δ¹⁸O *G. bulloides*
record of IODP Site U1387 (‰ VPDB) with MIS and substages indicated. B: Planktonic
foraminifera Summer SSTs (°C) from Site U1387 with the dashed line marking the late
Holocene level of 22°C. C: Alkenone derived SST record of IODP Site U1387 (magenta; Bajo
et al., 2020a and this study) in comparison to the Site U1385 record (dark grey; Rodrigues et
al., 2017; Rodrigues et al., 2020). D: The Azores Current factor (%) from IODP Site U1387
with the bar next to the scale indicating the modern range in Gulf of Cadiz surface samples
(Salgueiro et al., 2008). E: Thermocline water temperature (TWT) at Site U1385 (Bahr et al.,
2018; Bahr et al., 2017). Grey bars mark the odd-numbered MIS.

with those samples with lower similarity values (Fig. 5). The similarity between the respective
Site U1387 sample and the selected 10 modern analog database samples used to estimate the
temperatures is generally above 0.9. Some samples, often associated with extreme cold events,
have a lower similarity between 0.9 and 0.75 (Fig. 5D). At these specific lower similarity



samples, we observe a small contribution of "warm" species in a dominantly "cold" assemblage
leading to a non-analog situation. The species mix and consequently reduced similarity could
be linked to bioturbation and/or current transport in those contourite layers (Expedition 339
Scientists, 2013) or to the presence of *N. pachyderma* variants with different temperature
affinities (see discussion in subchapter 6.3).
The alkenone SSTs ($U^{k'}_{37}$-SST) record with annual temperatures ranging from 9.05 to
23.3 °C shows a clear glacial/interglacial cycle pattern (Fig. 6). Despite the differences in
amplitudes, both techniques registered the extreme cold events contemporaneously,
corroborating the interpretation of these results. The coldest period was recorded at the end of
MIS 22 between 870.3 and 864.3 ka. During this period, we recorded the highest percentage
of *N. pachyderma* (75.5 %) and the lowest temperatures with PF-SST of 4.8 °C for summer
and 0.9 °C for winter and $U^{k'}_{37}$-SST of 9.05 °C.

**6. Discussion**
**6.1 Persisting subtropical gyre influence**
The strong influence of the AzC in the region is evident through the comparison
between the modern planktonic foraminifera assemblage composition (Salgueiro et al., 2008;
Rufino et al., 2022) and the reconstructed assemblages from Iberian margin sediments (Girone
et al., 2023; Martin-Garcia et al., 2015; Salgueiro et al., 2010; Voelker et al., 2009). During the
EMPT, the interglacial and interstadial stages recorded the warmest temperatures, associated
with higher percentages of tropical, subtropical and transitional species (Fig. 3; 4D, E).
According to Salgueiro et al. (2008), high abundances of *G. ruber* (white), *T. sacculifer* and *G.*
*inflata* at the Iberian margin are indicative of the presence of the AzC's eastern branch (AzC
factor; Fig. 6D), whereas Storz et al. (2009) associated a much larger species group with the
AzC within the subtropical gyre (Table 1). The relative warm SSTs depicted throughout most
of the records, i.e. summer PF-SST within a range of 21 to 24.7 °C and winter PF-SST within
15 to 19.8 °C, is associated with the "AzC fauna" that include species from the transitional and
subtropical provinces (Salgueiro et al., 2008: Storz et al., 2009); so, we interpret those periods
with combined increased abundances of tropical, subtropical, and transitional species and with
values for the AzC factor above 30 % (Fig. 6D) as being under AzC and thus subtropical gyre
influence.
The persistent abundance of *N. incompta* (Fig. 3), slightly above the mean value of 18
% observed in the surface sediments (Salgueiro et al., 2008), also points to Portugal Current
contributions to the prevailing surface waters as it is a main contributor to the Portugal Current



factor (Salgueiro et al., 2008). Whereas higher abundances of *G. bulloides* during the EMPT
interglacial periods at western Iberian Margin Site U1391 (Fig. 1B) are interpreted as high
productivity upwelling periods (Singh et al., 2015), the same is not observed at Site U1387.
Here the percentages of *G. bulloides* generally remain below the local surface sediment mean
value of 34 % (Salgueiro et al., 2008) with no distinct glacial/interglacial variations, although
percentages increased during glacial MIS 24 and MIS 22 (Fig. 3). We interpret the *G. bulloides*
pattern at Site U1387 more as a temperature response, with limited influence of waters
upwelled in the major upwelling cell off Cape Saint Vicente (Fig. 1B) and advected towards
Site U1387. Nevertheless, the sporadic presence of *Chaetoceros* resting spores (diatoms)
within interglacial MIS 25e and MIS 28b (called MIS 27b in cited reference) document some
influence of seasonal upwelling at Site U1387 (Ventura et al., 2017). Interestingly, the rare
occurrences of planktonic foraminifera *N. dutertrei* in MIS 25e (Fig. 3) coincide with the
presence of the large-diameter marine diatom species *Coscinodiscus asteromphalus* (Ventura
et al., 2017), which can form large blooms and would thus be an ideal food source for *N.*
*dutertrei* (Schiebel and Hemleben, 2017).

The $U^{k'}_{37}$-SST data show similar patterns to the PF-SSTs with relatively stable

temperatures during interglacial and interstadial substages (Fig. 6). In contrast to the PF-SSTs,
the $U^{k'}_{37}$-SSTs (and abundance) reveal a clear cooling trend from the respective interglacial
optimum to the subsequent glacial maximum. This different pattern cannot solely be attributed
to the $U^{k'}_{37}$-SSTs reflecting annual mean temperatures instead of seasonal ones like the PF-
SSTs. During the last glacial maximum, the tropical and subtropical regions cooled (MARGO
project members, 2009; Osman et al., 2021; Tierney et al., 2020), so that we should expect a
similar climate sensitivity and cooling also during the EMPT glacial cycles, conform with the
$U^{k'}_{37}$-SST record of Site U1387 and other global records (McClymont et al., 2013; Naafs et al.,
2013; Rodrigues et al., 2017). The difference in the reconstructed SST pattern must, therefore,
be caused by the planktonic foraminifera fauna itself. While not obvious in the reconstructed
PF-SSTs, a decline in the abundance of the AzC species (Fig. 6D), largely driven by declining
*G. ruber* (white) contributions (Fig. 3), and a contemporary increase in Portugal Current
associated species (*G. inflata*, *N. incompta*; Fig. 3) is evident in all those interglacial-glacial
cycles. However, the AzC factor fauna (Fig. 6D) and other species linked to subtropical gyre
waters (Fig. 4D) retain relative high percentages, so that the transfer function is looking for
modern analogs in relative warm waters to estimate the EMPT faunal derived SSTs. Thus, due
to the faunas being too similar to modern subtropical gyre assemblages (Fig. 5D), the estimated
PF-SSTs at Site U1387 appear too warm and do not reflect the global cooling, also expected



for the North Atlantic subtropical gyre, during the transitions from the glacial inception to the
glacial maximum, at least for the glacial cycles covered by this study. The same pattern is also
evident for the PF-SSTs obtained for IODP Site U1385 (Martin-Garcia et al., 2015), which also
remained warmer than the corresponding $U^{k'}_{37}$-SSTs (Rodrigues et al., 2017). Site U1387 $U^{k'}_{37}$-
SSTs are ~2.5 °C warmer than the $U^{k'}_{37}$-SSTs of IODP Site U1385 on the southwestern
Portuguese margin (Fig. 1B), but the overall trends are the same (Fig. 6C). A similar
temperature difference between both sites is also visible for the PF-SST reconstructions for
MIS 21 to MIS 19, i.e. within the interval the records overlap (Martin-Garcia et al., 2015),
although the Site U1385 PF-SSTs were obtained using the artificial neural network method and
the original MARGO modern analog database from Kučera et al. (2005). We attribute the
temperature gradient to a stronger AzC influence at Site U1387, whereas Site U1385 is more
affected by the cooler Portugal Current waters (modern annual mean of 16.1 °C).

In the Gulf of Cadiz, the summer PF-SSTs and $U^{k'}_{37}$-SSTs reconstructed for the EMPT

interglacials were as warm as or slightly warmer than the current interglacial SST (~22 °C;
Salgueiro et al., 2014) in the case of the warmer interglacials, i.e. MIS 19c, MIS 21g and MIS
25e, and 1.5 °C cooler during MIS 23c and MIS 27 (Fig. 6B). However, neither of those
interglacial periods experienced surface waters as warm as during early Pleistocene interglacial
MIS 47, when $U^{k'}_{37}$-SSTs remained above 24 °C and subtropical planktonic foraminifera
abundance mostly above 40 % (Voelker et al., 2022).The warmest EMPT interglacial was MIS
21g, supported by high contributions of the subtropical+tropical fauna of up to 56.3 % *vs*. 45.5
% during MIS 19c and 38 % during MIS 25e (Fig. 4),  even though MIS 25e received the higher
amount of insolation (Rodrigues et al., 2017). The maximum percentages are comparable to
those observed during MIS 47 (generally exceeding 40 % and reaching up to 66.8 %), although
periods with such high contributions were much shorter during the EMPT interglacials. Much
of the subtropical+tropical fauna abundance is driven by the contribution of *G. ruber* (white)
(Fig. 3), which can attribute nearly half of the overall percentage. *G. ruber* (white) added less
to the MIS 23c fauna (11 %), but this interglacial had a unique fauna due to the higher influence
of subtropical species *G. siphonifera* (2.85 %) (Fig. 3) that persisted into MIS 23a when PF-
SSTs became less stable caused by the mixture of planktonic foraminifera provinces (including
subpolar and polar species).

Intra-interglacial SST variability is observed during several of the interglacials because

a cooling event was recorded in both SST reconstructions during MIS 21g, MIS 23c and MIS
27 leading to a three phased SST evolution, although the timing of the cooling event within the
interglacial period varied (Fig. 6). The PF-SSTs documented such a cooling event also for MIS



25e, where it occurred prior to the increase in the $U^{k'}_{37}$-SSTs in the younger phase of the
interglacial. As such the MIS 25e $U^{k'}_{37}$-SST pattern mimicked the one of MIS 11c on the
Portuguese margin (Rodrigues et al., 2011) and in the mid-latitudinal North Atlantic (Stein et
al., 2009), although on a shorter timescale.
Recently, Barker et al. (2021) proposed to treat MIS 28 as a "missed" interglacial and
we therefore include it in our comparison. The summer PF-SST and $U^{k'}_{37}$-SST records of Site
U1387 would support such a notion. Specifically, during interstadial MIS 28b, warm PF-SSTs
and $U^{k'}_{37}$-SSTs of 19.6 °C and 21.1 °C, respectively, and considerable contributions of the AzC
factor fauna suggest to categorize this period as an interglacial (Fig. 6).
Millennial-scale variability in the form of stadial/interstadial oscillations is observed in
our records, conform with evidence from other North Atlantic sites, evidencing significant
modifications in the North Atlantic's thermohaline circulation, the expansion of continental ice
sheets and sea ice, and the atmospheric circulation (e.g., Barker et al., 2021; Billups and
Scheinwald, 2014; Hernández-Almeida et al., 2015; Hodell and Channell, 2016; Hodell et al.,
2023a; Sun et al., 2021; Rodrigues et al., 2017). At Site U1387, one of the most dynamic
periods was the interval between MIS 25 and MIS 22, that points to highly variable surface
water conditions in the northern subtropical gyre region, in accordance with evidence from
DSDP Site 607 and IODP Site U1313 (Marino et al., 2008; Naafs et al., 2013). Here, we focus
on the interstadial periods, with the stadials being discussed later in subchapter 6.3. At Site
U1387, the interstadials recorded high mean summer PF-SSTs (20.8 °C) similar to the $U^{k'}_{37}$-
SSTs (20.0 °C), and a mean winter PF-SSTs of 15.7 °C. The warmest interstadial according to
the summer PF-SSTs was MIS 21e with 23.1 °C, whereas the cooler one reached around 19.9
°C (Fig. 5, 6). The interstadials had variable durations with MIS 22b being the longest period
with ~16 kyr and MIS 20b the shortest with ~1.5 kyr. Interstadial MIS 22b, occurring during
the middle of glacial MIS 22 and thus within the "900 ka event" period, had a summer PF-
SSTs in the general range of 19-21 °C, with the $U^{k'}_{37}$-SSTs being slightly cooler in the 17-18
°C range (Fig. 6). During the interstadials of MIS 23, MIS 22, MIS 21, and MIS 20 noteworthy
occurrences of the tropical, surface-dwelling species *T. sacculifer* are observed with 1.9 % on
average, but increasing to 2.9 % during MIS 21c (Fig. 3). The periods also registered the
greatest abundances of the subtropical species *G. falconensis* (average 2.7 %), increasing to 4.6
% during interstadial MIS 20b. Those indicator species, together with the higher AzC factor
fauna abundance (Fig. 6D), confirm prevailing subtropical gyre water influence and a strong
presence of the AzC at the southern Portuguese margin during the interstadials, in accordance
with previous observations on the southwestern margin (Girone et al., 2023; Martin-Garcia et





al., 2015; Singh et al., 2015). Those warm surface waters were subducted into the thermocline
levels and the subtropical North Atlantic Central Water (Bahr et al., 2018). Bahr et al. (2018)
link an intensified AzC, coupled to a strong Mediterranean Outflow Water, to their warmer
thermocline water temperatures, which is conform with our SST and faunal evidence (e.g., MIS
22b, MIS 21c, MIS 19a) (Fig. 6).

**6.2 *G. truncatulinoides* evidence for subtropical gyre circulation state**

*G. truncatulinoides* is a planktonic foraminifera species that prefers relatively warm,

nutrient-rich waters such as at the subtropical gyre margins (Ujiié et al., 2010; Rufino et al.,
2022). According to Kaiser et al. (2019) and Feldmeijer et al. (2015), the sinistral variant
dominates North Atlantic regions with a deep permanent thermocline, such as the central
subtropical gyre. Its presence at mid-latitudinal North Atlantic sites, especially during glacial
periods, can indicate the northward flux of subtropical waters and thus the position of the gyre's
northern boundary (Kaiser et al., 2019). In contrast, *G. truncatulinoides* (dextral) dominates in
the Atlantic's tropical waters. High percentages of that variant have been interpreted as
reflecting higher contributions of North Equatorial Current and Antilles Current waters to the
Gulfstream and thus enhanced westward and northward transport along the western boundary
of the subtropical gyre and into its central regions, corresponding with an enhanced gyre
circulation overall (Billups et al., 2020).

At Site U1387, the dextral variant dominates over the left coiling variant, but is only

present in relative low numbers (Fig. 4B, C), comparable to other sites in gyre boundary
locations (Kaiser et al., 2019). Nevertheless, the percentage contributions of *G.*
*truncatulinoides* to the Site U1387 EMPT faunas (Fig. 3) is in the same range as those observed
in surface sediments along the western Iberian margin, in the Gulf of Cadiz and the eastern
boundary current region off NW Africa (Salgueiro et al., 2008; Rufino et al., 2022). So, the
presence of *G. truncatulinoides,* especially in its right coiling form, supports subtropical gyre
influence during much of the studied interval, conform with the evidence discussed above. The
foremost characteristics of the *G. truncatulinoides* coiling record are, however, the % GTS
maxima during MIS 28, MIS 26, MIS 24, and stadial MIS 21b and a double peak during the
period from MIS 22a to MIS 21g (Fig. 4B, 7B). Many of those % GTS peaks have counterparts
at northern subtropical gyre Site 607 (Fig. 7D), where those maxima implicate the vicinity of
the gyre's northern boundary (Kaiser et al., 2019) and thus a gyre northward expansion not
much different from today, in agreement with the relative warm subsurface temperatures

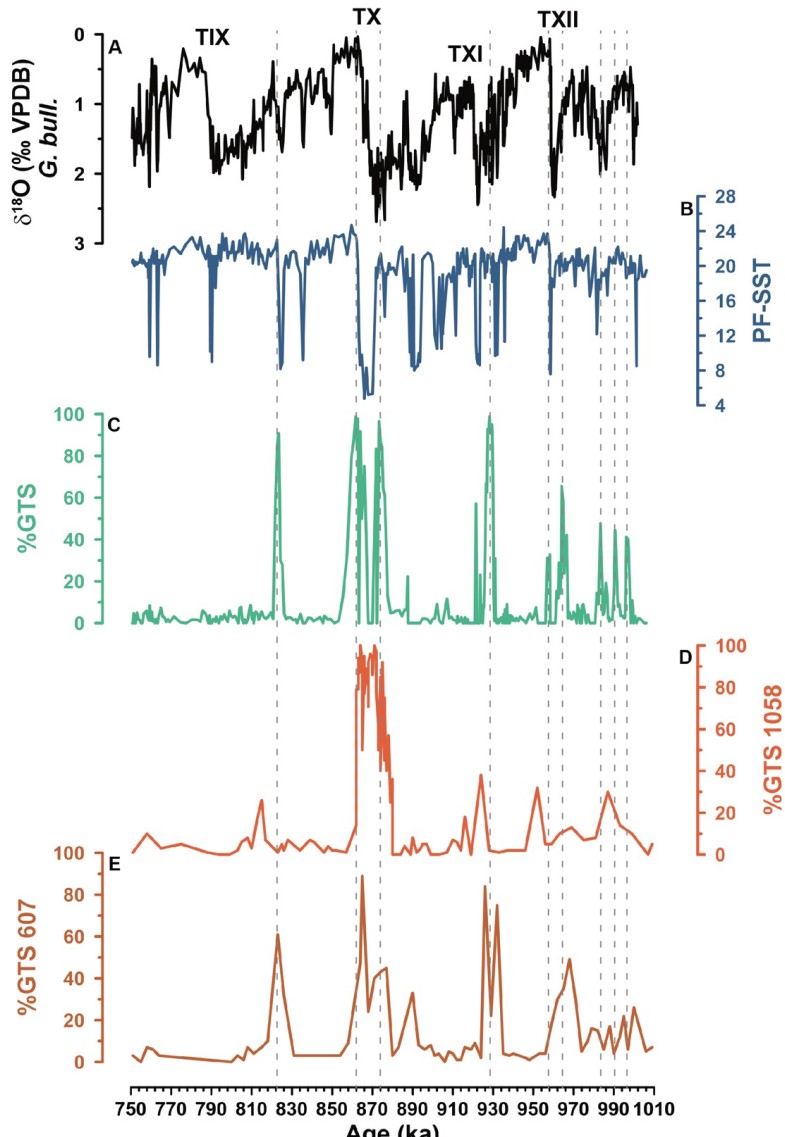

**Figure 7:** Subtropical gyre intensification episodes**.** A: $\delta^{18}O$ (‰) *G. bulloides* from IODP Site U1387. B: summer PF-SST from IODP Site U1387. C: Coiling ratio (%) of planktonic foraminifera *G. truncatulinoides* (sinistral) from IODP Site U1387. D: Coiling ratio (%) of planktonic foraminifera *G. truncatulinoides* (sinistral) from ODP Site 1058 (Kaiser et al., 2019). E: Coiling ratio (%) of planktonic foraminifera *G. truncatulinoides* (sinistral) from DSDP Site 607 (Kaiser et al., 2019). Dashed lines mark peaks of U1387 %GTS maxima. Terminations are indicated by the letter T and the respective Latin numerical.



reconstructed at the same location (Catunda et al., 2021). Since the total abundance of *G.*
*truncatulinoides* (sinistral) increased during most of those periods at Site U1387 as well (Fig.
4C), it is possible that the gyre circulation strength was comparable to late Holocene conditions
(Billups et al., 2020). Site U1387 recorded % GTS maxima during the MIS 28 and MIS 24 and
following the terminal stadial event (Hodell et al., 2015) of Termination XII (MIS 26/ MIS 25),
which have no counterparts at Site 607 (Kaiser et al., 2019), at least within the temporal
resolution and age model constraints (Fig. 7). Those maxima seem to indicate a vigorous
circulation in the eastern region of the subtropical gyre following an (extreme) cold event and
might be related to the subtropical gyre expanding northward again when the subarctic front,
the boundary between the subpolar and subtropical gyres, receded northward, but was still
mostly located south of 41°N, i.e. south of DSDP Site 607.
The most prominent feature in the % GTS records of Sites U1387, 607 and 1058 is the
period from MIS 22a to MIS 21g when % GTS temporarily reached values between 80 to 100
% (Fig. 7). At western boundary/Gulf Stream ODP Site 1058 the feature is one long lasting
(~20 kyr) peak, whereas both at DSDP Site 607 and IODP Site U1387 a double peak is
observed. Based on their Sites 1058 and 607 data, Kaiser et al. (2019) posited that the North
Atlantic's subtropical gyre expanded as far north as 41°N (or further) during glacial MIS 22a
and that its circulation was more vigorous than during the last glacial maximum. This scenario
agrees with the warm subsurface temperatures reconstructed at IODP Site U1313 during MIS
22a (Catunda et al., 2021), which were not much cooler than interglacial levels and indicate an
expanded layer of subtropical gyre waters. The Site U1387 record now corroborates an
expanded and strong subtropical gyre with evidence from the gyre's eastern boundary, i.e., the
gyre's eastern boundary must have been located in the vicinity of Site U1387. During the
terminal stadial event of Termination X, the subtropical gyre contracted in the north and east
leading to the % GTS minima at Sites 607 and U1387 (or even temporary absence of the species
at U1387), whereas its western boundary remained near the position of Site 1058 (Kaiser et al.,
2019). When the subarctic front receded northward after the terminal stadial event of
Termination X, the subtropical gyre expanded again as evidenced by the % GTS maxima at
Site 607 and U1387 (Fig. 7), facilitating subtropical water transport to the north and deep-water
convection in the North Atlantic (Fig. 8G) (Hodell and Channell, 2016; Hodell et al., 2023a;
Kaiser et al., 2019) and the establishment of interglacial conditions.





**6.3 The extreme cold events**


Site U1387 recorded several short stadial events (~2 kyr) following the glacial
inceptions and as terminal stadial events with winter PF-SSTs dropping to ~5 °C during MIS
24a or even to freezing temperatures of 0 °C during MIS 22a (Fig 5, 8B). The $U^{k'}_{37}$-SSTs during
those terminal stadial events also reflect extremely low temperatures, but only reaching 10 °C
during MIS 22a and MIS 24a (Fig 6C). The southern position of the subarctic/Arctic front
during those stadial periods (Martin-Garcia et al., 2015; Rodrigues et al., 2017), facilitated the
presence of the polar species *N. pachyderma,* which reached between 80 % (MIS 22a) and 50
% (MIS 24a) (Fig 8C), as well as a general increase in the number of polar and subpolar species
(Fig 8D). The high percentages of *N. pachyderma* are much higher than those observed during
the Heinrich events of the last glacial cycle in the Gulf of Cadiz (< 20 %) and also exceed those
observed in general on the southwestern Portuguese margin during the last 400 kyr (<40 %)
(Salgueiro et al., 2014; Salgueiro et al., 2010; Singh et al., 2023; Voelker and De Abreu, 2011).
They drive the extremely cold SST estimated for the PF-SST, which appears to introduce a
"cold bias" for the winter PF-SST that are much colder than the (annual mean) $U^{k'}_{37}$-SSTs (Fig.
6, 8). The *N. pachyderma* morphotypes (supplementary figure 1) are similar to those found
today in the subpolar to polar North Atlantic and those observed in contemporary "middle"
Pleistocene sediments in the Alboran Sea (western Mediterranean Sea) (Serrano and Guerra-
Merchán, 2012). Although *N. pachyderma* adapted to the colder, polar conditions 1100-1000
kyr ago (Huber et al., 2000; Kucera, 2007), thereby establishing the modern polar *N.*
*pachyderma* variant (genotype Ia), a recent review of genetic diversity in planktonic
foraminifera from the modern global ocean (Morard et al., 2024) revealed that other *N.*
*pachyderma* genotypes occur only in lower to mid-latitudinal waters of the Atlantic (e.g., Va,
VIa), whereby genotype VIa is well established in the mid-latitudinal North Atlantic, especially
in the AzC region, and the Mediterranean Sea. Serrano and Guerra-Merchán (2012) postulated
that their early Pleistocene *Neogloboquadrina* specimens from the Alboran Sea might include
two groups with different temperature affinities, one being the modern polar variant and the
other living in warmer waters and/or upwelling conditions. Their observations indicate that the
mid-latitudinal North Atlantic genotype VIa or a precursor of it might have already been
present in the early Pleistocene. As it is difficult to distinguish between the genotypes based
on morphology, it is possible that the Site U1387 EMPT *N. pachyderma* specimens include
both the polar and the mid-latitudinal, warmer water affinity variants. Presence of a warmer
water variant would agree with the low, but noticeable contemporary presence of various
subtropical species and of tropical species *T. sacculifer* in the Site U1387 faunas (Fig. 3, 4D)

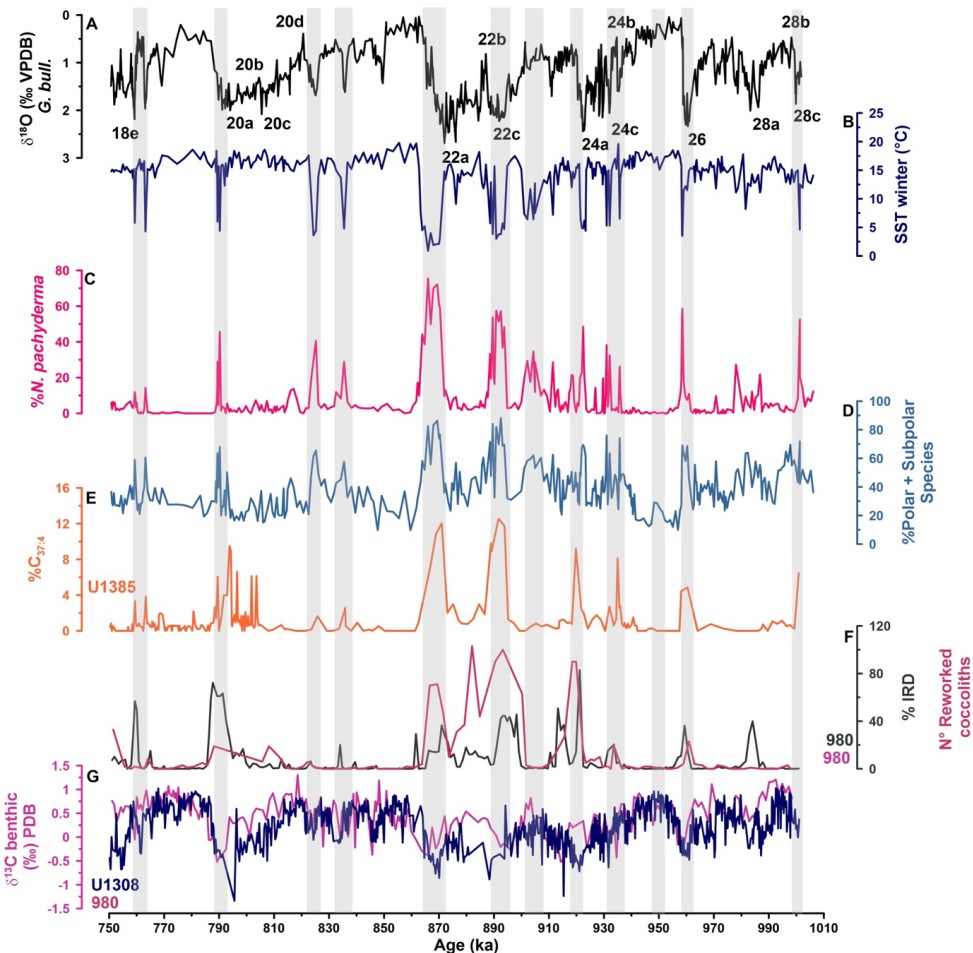

**Figure 8:** The extreme cold events. A: IODP Site U1387 *G. bulloides* $\delta^{18}$O (‰ VPDB) record with MIS and substages. B: Winter PF-SST (°C) from IODP Site U1387. C: Abundance (%) of the planktonic foraminifera *N. pachyderma* from IODP Site U1387. D: Abundance (%) of polar and subpolar species at IODP Site U1387. E: $\%C_{37:4}$ freshwater indicator from IODP Site U1385 (Rodrigues et al., 2017). F: Number of reworked coccoliths (Marino et al., 2011) and abundance (%) of ice-rafted debris from ODP Site 980 (Wright and Flower, 2002). G: $\delta^{13}$C benthic foraminifera (‰ VPDB) (magenta line) from ODP Site 980 (Wright and Flower, 2002), and $\delta^{13}$C benthic foraminifera (‰) (dark blue line) from IODP Site U1308 (Hodell and Channell, 2016). Gray bars mark cold events.

that hint to some AzC/subtropical gyre influence. This is especially true for MIS 22 when such subtropical water contributions would be consistent with the relative northern expansion of the



subtropical gyre (Catunda et al., 2021; Kaiser et al., 2019). Because the modern analog
technique used to calculate the PF-SST relies on the percentage contributions of *N. pachyderma*
to the total fauna it looks for modern analogs in the Nordic Seas and Labrador Sea and therefore
overestimates the cooling, if the % *N. pachyderma* values include relevant contributions of a
warm water variant or where that variant dominates. So, for the interpretation of the cold stadial
events we give more weight to those events where PF-SSTs and $U^{k'}_{37}$-SSTs show contemporary
cooling and caution that some of the extreme cold PF-SSTs might be overestimated.

The terminal stadial events (Fig. 8) are all clearly marked in the PF-SSTs and $U^{k'}_{37}$-

SSTs record with extreme cooling. The Termination X event (MIS 22a) lasted the longest (6
kyr) and was the coldest, registering with lowest SSTs of the whole study interval. In contrast
to the southwestern Portuguese margin records from Site U1385 (Girone et al., 2023; Rodrigues
et al., 2017), southward incursion of cold surface waters to Site U1387 during Termination IX
was much more limited as indicated by the diminished cooling in regard to amplitude and
duration. When compared to the others, an atypical terminal stadial event occurred at the end
of MIS 28a with low *N. pachyderma* abundances (20 %) and relatively warm PF-SST (10 °C)
(Fig. 8). The event presents, however, an assemblage dominated by polar and subpolar species
(60 %) and evidence of ice-rafting at ODP Site 980 in the subpolar North Atlantic (Fig. 8F)
(Wright and Flower, 2002). All terminal stadial events registered at Site U1387 coincided with
ice rafting and melting icebergs in the North Atlantic (Fig. 8E, F) (Hodell and Channell, 2016;
Marino et al., 2011; Rodrigues et al., 2017; Wright and Flower, 2002) and a related strong
reduction of the AMOC depth as evidenced by the presence of AABW in water depths normally
occupied by NADW (Fig. 8G) (Hernández-Almeida et al., 2015; Hodell and Channell, 2016;
Hodell et al., 2023a). The Site U1387 records, therefore, provide further evidence for an
extreme contraction of the subtropical gyre in the eastern North Atlantic during those events
and indicate that the subarctic front advanced much further south during those events than
during the Heinrich events of the last glacial cycle or any terminal stadial events of the last 400
kyr, as already previously suggested by Rodrigues et al. (2017).

In addition to the terminal stadial events, there occurred stadial events during MIS 24c,

MIS 23b, MIS 21d, MIS 21b, MIS 19b, and MIS 18e with similar environmental characteristics
(Fig 8). Although those periods presented lower % *N. pachyderma* between 20 % during MIS
18e and 40 % during MIS 24c, the assemblages were dominated by polar and subpolar species
(60-80%) resulting in very cold winter PF-SSTs. The transition to the glacial maximum of MIS
24 was marked by three stadial/interstadial oscillations, with the first two occurring early on
and with only 2 kyr separating them. The last stadial was a little cooler (4.9 °C) but was





associated with a strong increase in *N. pachyderma* abundance (50 %) and high amounts of
IRD (80 %) and reworked coccoliths (90 %) in the subpolar North Atlantic at the ODP Site
980 (Marino et al., 2011; Wright and Flower, 2002) (Fig. 8). This evidence, combined with the
lower $U^{k'}_{37}$-SST (10 °C) at Site U1387 and the presence of freshwater input at Site U1385
(Rodrigues et al., 2017), indicate a strong southward displacement of the subarctic front also
during this event. The cold events in MIS 23b, MIS 21b and MIS 18e were associated with
hardly any cooling in the $U^{k'}_{37}$-SSTs (Fig. 6), representing potential cases of "cold bias" in the
PF-SSTs.
The transition between MIS 23 and MIS 22 initiated the "900 ka event". Cooler
temperatures during MIS 23 led to an abrupt increase in Antarctic ice volume and thus lowering
of the sea level to 120 m below present (Elderfield et al., 2012).  The lower sea level permitted
the advance of marine-based ice sheets around the North Atlantic with impacts on ice-rafting
and subarctic front movements (Hodell and Channell, 2016). At Site U1387, those background
conditions resulted in a cooling event during MIS 23a that is clearly visible in the planktonic
foraminifera records, but not the $U^{k'}_{37}$-SST records of either Site U1387 or Site U1385 (Fig. 4,
6, 8). It is, however, contemporary with a short cooling in the subtropical gyre's subsurface
waters at Site U1313 (Catunda et al., 2021). The cooling trend initiated with this event
culminated in the first, prolonged period of extreme cold conditions during MIS22c at Site
U1387 (Fig. 6, 8). In the mid-latitudinal North Atlantic ice-rafting and iceberg melting (Hodell
and Channell, 2016; Marino et al., 2011; Wright and Flower, 2002) led to freshening of the
surface waters, even as far south as Site U1385 (Fig. 8E) (Rodrigues et al., 2017), and
subsequently to a reduction in the AMOC depth (Fig. 8G) (Hernández-Almeida et al., 2015;
Hodell and Channell, 2016; Hodell et al., 2023a; Wright and Flower, 2002). The associated
contraction of the subtropical gyre is also reflected in the subsurface waters at Site U1313
cooling by 2 °C to the range of 4 °C (Catunda et al., 2021), a cooling that is not seen during
MIS 22a when the subtropical gyre was stronger (Kaiser et al., 2019).

**Conclusions**
The planktonic foraminifera faunal and SST records of IODP Site U1387 revealed that
subtropical gyre waters, especially those related to the AzC, greatly influenced the Gulf of
Cadiz during the EMPT interval from MIS 28 to MIS 18, even during the transitions to full
glacial conditions following the glacial inceptions. The planktonic foraminifera fauna includes
species from all four provinces with the subpolar and polar species dominating during the
extreme cold events, in particular the terminal stadial events. The faunal diversity differed



slightly from the Holocene, a topic which implications for ecosystem state and restoration efforts in the region will be explored further in the future. Interglacial periods and several of the interstadials experienced SST as warm or slightly warmer than today and registered persistent AzC influence. The warmest interglacial period was MIS 21g and the coolest, as to be expected from the global climate state, MIS 23. MIS 23 exhibited a particular subtropical planktonic foraminifera fauna, which, in contrast to the other interglacials, included a lesser contribution of *G. ruber* white, but higher ones of *G. falconensis* and *G. siphonifera*. Interestingly, tropical species *T. sacculifer* was present in low percentages throughout MIS 23 and even glacial MIS 22, which included the two periods with the coldest SST of the studied time interval.

Glacial MIS 22a with the terminal stadial event of Termination X stands out as a special time. On the one hand, the highest % *N. pachyderma* and coldest SST during a prolongated period indicate extreme cooling and incursion of subpolar waters into the latitudes of the Gulf of Cadiz. This is only possible if the subarctic front was shifted southward in the eastern North Atlantic. On the other hand, Kaiser et al. (2019) infer the subtropical gyre expanded at least as far north as 41°N, with a circulation more vigorous than during the last glacial maximum. The % GTS data of Site U1387 agrees with such a scenario and indicates that the eastern boundary of the subtropical gyre was in the vicinity of the Gulf of Cadiz during much of MIS 22a, with the exception of the peak of the terminal stadial event. As such, the Gulf of Cadiz is once again confirmed as an important confluence region during glacial periods.

Millennial-scale climate variability is clearly recorded as stadial/interstadial SST oscillations during the transition from MIS 25 to MIS 24, whereas during the MIS 21 to MIS 20 transition only MIS 21d and MIS 21b experienced short-term cooling events. Likewise, MIS 19b was associated with a short-term cooling event.

By combining evidence from planktonic foraminifera assemblages with two types of SST reconstructions, we have identified potential biases in our reconstructions. The persistent presence of subtropical gyre and AzC related species in our samples leads to overestimated PF-SSTs following the glacial inceptions and during part of the glacial periods, which becomes obvious in comparison with the $U^{k'}_{37}$-SSTs. On the other hand, the "hidden" presence of a *N. pachyderma* variant with an affinity to warmer AzC current waters, mixed in with the polar variant, probably leads to a cold bias in the PF-SST reconstructions of the extreme cold events. The temporal evolution of *N. pachyderma* and its affinities and potential implications for paleoceanographic reconstructions in the region will be explored in the future, when the Site U1387 planktonic foraminifera faunal records going back to 1500 ka have been completed.



**Data availability.** *G. bulloides* oxygen isotope and $U^{k'}_{37}$ SST raw data for MIS 21-MIS 26 were already published in Bajo et al. (2020a); note that Bajo et al. (2020b, c) ages listed in Pangaea differ from the age model used in this manuscript:

https://doi.pangaea.de/10.1594/PANGAEA.914401

https://doi.pangaea.de/10.1594/PANGAEA.914400

Data published in this manuscript will also be archived at Pangaea (to be submitted as soon as the preprint is published and the manuscript has a doi) and are currently provided as supplementary material for the review process.

**Supplement.** The supplement related to this article is available online at:………..

**Author contributions.** AHLV initiated and designed the study and secured funding for the biogeochemical and stable isotope analyses. AM produced the planktonic foraminifera faunal data and, together with AHLV, wrote the first draft of the manuscript. ES trained AM in the application of SIMMAX and the interpretation of its results. MP produced the lipid biomarker data under the supervision of TR who also made the final quality control of the results. HK performed the stable isotope analyses at MARUM. All authors, with the exception of MP, who left science in the meantime, read and commented on the draft of the manuscript and approved its final version.

**Competing interests.** At least one of the (co-)authors is a member of the editorial board of *Climate of the Past*. The authors have no other competing interests to declare.

**Acknowledgments**. The samples for this study were provided by the Integrated Ocean Drilling Program (2003-2013) and we thank the Bremen Core Repository and its staff for support in sampling the sections. We thank W. Soares, Cremilde Monteiro and the research fellows contracted by the MOWCADYN project for help in preparing the samples in the micropaleontology and sedimentology lab at IPMA.

**Financial support.** The stable isotope and lipid biomarker analyses were made possible through Fundação para a Ciência e a Tecnologia (FCT) funded R&D project MOWCADYN (PTDC/MAR-PRO/3761/2012). Additional financial support from FCT was provided to the Centro de Ciências do Mar do Algarve (CCMAR) through projects: basic funding




UIDB/04326/2020 (https://doi.org/10.54499/UIDB/04326/2020); programmatic funding
UIDP/04326/2020 (https://doi.org/10.54499/UIDP/04326/2020) and the CIMAR associated
laboratory funding LA/P/0101/2020 (https://doi.org/10.54499/LA/P/0101/2020). A. Mega is
funded by FCT/CCMAR through PhD fellowship CCMAR01/UIDP/04326/2020. A. Voelker
acknowledges her Investigador FCT grant (IF/01500/2014), which provided her salary during
the initial phase of the study (2015-2021). The SEM work was made possible through access
to the GOLD lab facilities at IPMA funded by the EMSO-PT infrastructure project (POCI-01-
0145-FEDER-022157).

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
