# Peer review of "The Early Middle Pleistocene Transition in the Gulf of Cadiz (NE Atlantic) an"

_EGUsphere, 2024_

## Author Comment (AC2)

Review of **Subtropical gyre persistence in the Gulf of Cadiz, southern Iberian margin, interrupted by extremely cold surface water incursions during the Early – Middle Pleistocene Transition** by Mega et al.

We thank the reviewer for the reading our manuscript and providing critical feedback that will help us to improve our manuscript. Our responses to the specific comments are given in blue.

This study presents new foraminiferal faunal and alkenone measurements, from a site off the Iberian Margin, to identify changes in surface ocean temperature and hydrography across the MPT. The data presented add a valuable resource for improving our understanding of an important climatic transition and ultimately should be published in a venue such as this. Overall the data are well presented and much of the discussion is sensible. However, I am not so confident about the use of coiling direction in G. truncatulinoides as a gyre indicator and I think more or different discussion on surface versus deep circulation changes needs to be included.

More specifically, the authors make use of the trunc coiling direction proxy to support their arguments yet this approach appears quite speculative and gives inconsistent results (e.g. MIS 1 versus MIS 11c according to Billups et al., 2020). Furthermore the proxy has not been demonstrated in the region under study here and it is not at all clear that the %GTS record presented here can be interpreted in terms of gyre strength. I note that the absolute counts of trunc are generally very low (Fig. 4C, typically <10). Kaiser et al., 2019 suggested that low counts should not be relied on and considered only those samples with >20 trunc counts.

The %GTS proxy is a new one and our study is the first to apply it outside of the group of Katharina Billups. The proxy has not been demonstrated for the eastern boundary region of the North Atlantic subtropical gyre but when defining the proxy Billups et al. (2016) took evidence from the complete North Atlantic into account (see their figure 2). The patterns observed by Billups et al. (2016) is confirmed by the updated map below (Figure 1), which is using the % *G. truncatulinoides* right and % *G. truncatulinoides* left data from the Salgueiro et al. (2014) modern analog database to calculate %GTS (keeping in mind that total *G. truncatulinoides* count numbers in the surface samples might not fulfill the cut-off criteria defined by Kaiser et al. (2019); we will include this map in the supplementary material of our revised manuscript as it provides evidence for the eastern boundary current region). The oceanographic conditions in our study region are strongly linked to the eastern boundary circulation of the North Atlantic subtropical gyre and has, in addition, through the Azores Current a direct link to the Gulf Stream. The Azores Current influence on the southern and southwestern Portuguese margin is clearly reflected in Figure 1 by the generally mid-range %GTS values. So, we are confident that the proxy can be applied in our study region.

From the planktonic foraminifera species composition on the Portuguese margin we know that subtropical gyre circulation was not exactly the same during MIS 1 and MIS 11c, because deep dwelling tropical species *G. menardii* and *S. dehiscens* are present in samples from MIS 11c (and MIS 9e; e.g., Voelker et al. 2010) but are not found in MIS 5e or MIS 1 samples (A. Mega, unpublished data). So, we would argue that the differences observed between MIS 1 and MIS 11c are not inconsistencies but changes in the (subsurface?) subtropical gyre that we actually have to explore further (and based on a wider site distribution) in the future.

Figure 1: % GTS pattern in the North Atlantic calculated based on the % *G. truncatulinoides* left and % *G. truncatulinoides* right data available in the Salgueiro et al. (2014) surface sample database; stations with just a total 0.1% *G. truncatulinoides* (left and right combined) contributions to the total fauna were excluded.

[Figure]

We thank the reviewer for the comment on the absolute counts! After verification (also with the data uploaded to PANGAEA) it became obvious that Figure 4C was not showing the total counts of *G. truncatulinoides*, but the numbers of *G. truncatulinoides* left only! This was now been corrected and is shown in the updated version of Figure 4 below (as Figure 2).

Since the count numbers are low during the %GTS minimum during the overall peak at the MIS 22/MIS 21 transition we need to revise the text in our discussion and exclude this part. Also, the peak at 918 ka will have to be excluded. We intend to update Figure 7 and show there for U1387 just the reliable %GTS peaks, making it easier for the reader to compare the records, because with the exclusion of that central section of low values, the overall pattern at Site U1387 becomes much more similar to the data of Sites 1058 and 607. We thank the reviewer for reminding us to keep the count numbers in mind, which we had overlooked in that part!

The record of %GTS presented here also shows an inconsistent link with glacial/interglacial climate, with high values observed during both warm and cold periods (although more dominantly during cold periods). And high numbers of shells are only observed when GTS is high, which could mean that the thermocline is deep (gyre is healthy?) only during those times.

As pointed out by the reviewer many of the %GTS maxima coincide with stadial phases of the millennial-scale climate variability and the gyre strengthening is probably related to the

contraction of the gyre. Only the start of MIS 21g and the first peak during MIS 28b, which is, however, less reliable because the two higher % points are based on count numbers at the cut-off limit or just below it, coincide with warmer climate phases. Higher numbers of *G. truncatulinoides* are not only associated with high % GTS as now evident from the correct data plotted in Figure 2 (revised Figure 4). In the manuscript figure panel 4C was actually showing the number of GTS; so, based on that the reviewer's statement is correct.

Figure 2: Revised version of manuscript figure 4 showing the total number of *G. truncatulinoides* specimens counted in panel C.

[Figure]

Yet the authors seem to suggest that the dominance of both right and left coiling varieties can be taken as evidence for a strong /healthy subtropical gyre. For example, (L570) "the presence of *G. truncatulinoides,* especially in its right coiling form, supports subtropical gyre influence during much of the studied interval" and later, (L613) "When the subarctic front receded northward after the terminal stadial event of Termination X, the subtropical gyre expanded again as evidenced by the % GTS maxima at Site 607 and U1387 (Fig. 7), facilitating subtropical water transport to the north and deep-water convection in the North Atlantic (Fig. 8G) (Hodell and Channell, 2016; Hodell et al., 2023a; Kaiser et al., 2019) and the establishment of interglacial conditions." I think I am not mistaken in assuming that some sort of the subtropical gyre will always exist but with variations in position and intensity. Yet I cannot see how the results presented here help with understanding those variations and the authors' discussion offers little illumination. I am sorry that my words seem harsh but I have read the Billups and Kaiser papers and read through the text more than once and still am left wondering what I have learned from the record %GTS.

The statement in Line 570 is related to *G. truncatulinoides* as a subtropical species in general. As shown in Figure 1 the study region has both left and right coiling specimens in the surface sediments. We will delete the "especially in its right coiling form" phrase in the revised manuscript to avoid confusion for the reader. We agree with the reviewer that some form of subtropical gyre will always have existed. The %GTS data offers additional evidence on North Atlantic subtropical gyre conditions and we don't want to suppress that evidence, even if parts of the scientific community like the reviewer might not (yet) see its validity. We hope with including Figure 1 in the supplementary material and with revising the discussion text for the Termination X interval that the evidence and text becomes clearer.

Cited references:
Salgueiro, E., Naughton, F., Voelker, A. H. L., de Abreu, L., Alberto, A., Rossignol, L., Duprat, J., Magalhães, V. H., Vaqueiro, S., Turon, J. L., and Abrantes, F.: Past circulation along the western Iberian margin: a time slice vision from the Last Glacial to the Holocene, Quaternary Science Reviews, 106, 316-329, 10.1016/j.quascirev.2014.09.001, 2014.

Voelker, A. H. L., Rodrigues, T., Billups, K., Oppo, D., McManus, J., Stein, R., Hefter, J., and Grimalt, J. O.: Variations in mid-latitude North Atlantic surface water properties during the mid-Brunhes (MIS 9-14) and their implications for the thermohaline circulation, Clim. Past, 6, 531-552, doi: 10.5194/cp-6-531-2010, 2010.

Some of the results section is overly descriptive and potentially verbose. Consider condensing to save space and improve readability.

We will condense the results section in the revised version.

Below I list the major items for consideration, followed by more technical comments.

Line 40/41: What is the evidence for this sequence of events? You imply that extreme AMOC weakening caused the migration of the subarctic front but could it not be the other way around?

There is no concrete evidence per se as the age model errors from the various IODP Sites in the North Atlantic do not allow to deduce a true sequence of events. Ice-rafting event and evidence for a shallower AMOC, i.e., lower benthic $\delta^{13}C$ values, are contemporary at IODP Site U1308. However, the statement in the abstract is written from the perspective of the southern Portuguese margin, meaning the subarctic front was displaced (assuming an interglacial position similar to today) from a latitude of 51°N (or even further north in the eastern basin) to a latitude of 37°N. In our opinion, such an extreme displacement is not possible just by (initial) freshening in the subpolar gyre and the associated southeastward displacement of the subarctic front. One needs the "amplification" of a reduced overturning circulation and associated reduced heat and salt flux to the more northern latitudes (i.e., a weakened and/or "displaced" North Atlantic Current) to push the subarctic front as far south as the latitudinal range of the southern Portuguese margin.

L53: We don't yet know what caused the transition to more intense and longer-lasting glacials although $CO_2$ and ocean circulation changes were intimately involved. Perhaps tone down the implied certainty of cause and effect here.

We will change the wording to avoid implying certainty.

L56: A recent paper by Hines et al. [*Hines et al.*, 2024] suggests that no substantial change in glacial deep ocean circulation occurred across the MPT. This requires us to rethink some of our assumptions about ocean dynamical changes across the MPT and should be cited (and discussed – see later).

We will add Hines et al. (2024) as additional reference and check if there are instances where their suggestions might be included in the discussion later on.

L93: Again, there is emphasis placed on changes CAUSED by AMOC variations across the MPT but this order of cause and effect needs to be reconsidered.

We will delete the phrasing of "a result" and adjust the wording to be more open/descriptive.

L140: The study relies heavily on the use of trunc coiling direction as a proxy for gyre circulation as described by Billups et al., 2016 (which I think implies a weaker gyre circulation during the LGM). A recent study by Wharton et al., [*Wharton et al.*, 2024], using sites in a similar region as the Billups study, suggests that the glacial gyre was stronger and deep than today. This paper needs to be considered in the light of how reliable the trunc proxy can be.

The %GTS proxy is just a small part of all the new data generated for the current study and it therefore does not rely "heavily" on it. However, since it is a new proxy, we will add the cautionary note that there might exist inconsistencies with other reconstructions (e.g., Wharton et al., 2024) that need to be explored further in the future (which is outside of the scope of the current manuscript).

L218: Please give details of how the age model used here differs from the Bajo study. The Bajo study was defined by it's age model (tied to U1385 and speleothem records) and changing it here would seem to contradict the Bajo study.

This comment does not make much sense for our manuscript because the Bajo et al. study and thus age model does not cover the complete interval presented here. The tuning to the speleothem chronology also led to some extreme changes in the sedimentation rates. We will add a figure in the supplementary material to show the age differences between the Bajo et al. speleothem ages and the LR04/Probstack ages. We have never compared the two age model approaches (U/Th dated vs. orbitally tuned), but do not expect major differences. Similar to the Site U1385 record, our tuning target, the Site U1387 timeseries is much longer than presented here (MIS 17 to MIS 52 for the multi-proxy data) and we therefore established a new age model that allows placing the complete timeseries on a consistent LR04/Probstack chronology.

L241: "the amount of sinistral coiling direction of this species increases when the subtropical gyre circulation is more intense." Specifically, a higher concentration of left coiling truncs supposedly reflects a deeper permanent thermocline and hence healthy gyre circulation. But this depends very much on the site selected. Are the authors confident that their site will work in the same way?

When defining the parameter Billups et al. (2016) used data from the whole North Atlantic - see their figure 2. Our site is located on the eastern edge of the subtropical gyre and is strongly influenced by the Azores Current and the Portugal-Canary Currents which represent the eastern boundary re-circulation of the subtropical gyre. The boundary distinguishing between left and right coiling dominance in the Northeast Atlantic in figure 2 of Billups et al. (2016) actually follows the northern boundary of the Azores Current and if confirmed in the updated %GTS map shown in Figure 1. So, yes, we have confidence that the proxy should work in our region and it was tested here for the first time. The coincidence of %GTS peaks at Site 607, 1058 and U1387 (our figure 7) seems to confirm this (initial) assumption.

L465: I wouldn't disagree with cooling bit but I don't think we know what the climate sensitivity would be.

We will adjust the wording here to make clear that we mean this statement in a general sense of "a cooling trend" to be expected and not that climate sensitivity during the LGM was the same as during the MPT glacials.

L476: so are the waters at the site actually warmer or is it just an illusion? And if so, then should you plot the faunal SST at all?

Based on the faunal evidence the waters were warmer than, for example, many of the late Pleistocene glacial periods. The higher percentage of subtropical species is creating the "bias", but also indicates that the waters were relative warm and contained a relevant contribution of subtropical gyre waters. The questions are just how warm or how gradual the cooling was at the zooplankton level; questions which might only be solved by Mg/Ca or clumped isotope temperature reconstructions as the lipid biomarker SST might include their own caveat (e.g., Ausin et al., 2021), which we did not discuss in the manuscript as the focus is on the foraminifera faunal data. In our opinion, it makes sense to show the faunal SST -not only to highlight the probable bias, which also affects other studies in the region (e.g., Martin-Garcia et al., 2015), but also because some "truth" is hidden in the data, which should not be ignored.

Ausín, B., Magill, C., Haghipour, N., Fernández, Á., Wacker, L., Hodell, D., Baumann, K.-H., Eglinton, T.I., 2019. (In)coherent multiproxy signals in marine sediments: Implications for high-resolution paleoclimate reconstruction. Earth and Planetary Science Letters 515, 38-46, https://doi.org/10.1016/j.epsl.2019.03.003.

L492: But Uk37 temp was as high during MIS 27 as e.g. MIS 21.

We will adjust the wording here to make clear that the cooler MIS 27 statement refers to the fauna derived SST estimates.

L507: earlier (L458) you stated that interglacials were associated with stable SSTs

We thank the reviewer for pointing out this inconsistency in our text. We will adjust the wording in line 458.

L512: the difference between faunal and Alk SST is important and should be further discussed (is it to do with the gyre again?). Also the comparison with MIS 11 is interesting but needs to be filled out a lot.

We will add a bit more text here but will not go for a full comparison with MIS 11, which is beyond the scope of the current manuscript. The early phases of MIS 25e and MIS 11c were both associated with the evidence for (more persistent) upwelling. So, the difference is more caused by ecological preferences of the studied phyto- (coccolithophores) and zooplankton (foraminifera) groups than the gyre circulation.

L515-519: this is interesting, since MIS 28b was only missed at more northerly sites and was warm at the southern margin of the Atlantic Inflow and at the site described here. Could the authors comment on the conclusions of Barker et al. (2021) on how surface ocean circulation might have changed across the MPT?

Since it is out of the scope of the current manuscript, we will not comment on the conclusions of Barker et al. (2021) in it. From Site U1387 we have now faunal evidence, in particular related to *N. pachyderma*, that MIS 29/MIS 28 (prior to the data presented in the current manuscript) might have been a critical period in species variant evolution in the mid-latitudinal N Atlantic, in agreement with the general 1.1-1 Ma time frame of Huber et al. (2000) for the emergence of the polar variant of *N. pachyderma.* As the Barker et al. (2021) study relies soly on % *N. pachyderma* evidence one might have to be more careful in the data interpretation, especially at sites under subtropical gyre (607) and North Atlantic Current (U1304, ODP 980/981) influence. It would probably also be relevant to look into the full planktonic foraminifera fauna and not just the % *N. pachyderma* to obtain a more complete picture of circulation changes/inflow strengthening. We might get back to this topic when the manuscript on the Neogloboquadrinids evolution at Site U1387 will be written (hopefully later this year).

L688: Benthic $d^{13}C$ cannot be interpreted as a quantitative proxy for the AMOC.

The modeling work of Muglia and Schmittner (2021) shows that benthic $\delta^{13}C$ is reflecting more overturning depth and not strength. That is the reasoning we are following with our wording here. We will add the Muglia and Schmittner reference in the revised manuscript to make this clearer.

Muglia, J., Schmittner, A., 2021. Carbon isotope constraints on glacial Atlantic meridional overturning: Strength vs depth. Quaternary Science Reviews 257, 106844, doi: https://doi.org/10.1016/j.quascirev.2021.106844.

L711: this was speculation in the Elderfield paper

We will adjust the wording to point out that this was a speculation.

Technical:

We will address all the technical issues in the revised manuscript.

L19: Cooler SSTs in general or during glacials?

L110: Never proven – replace with demonstrated?

L202 etc: Use of 'weighted' instead of 'weighed'

L322: "In total, 16 species were identified (Table 1; Fig. 3), with the diversity of the subtropical fauna appears to be diminished" – please check sense (or lack of)

Para beginning L352: This is very descriptive and wordy. Please condense.

L389: do you mean extremely short or extremely cold?

L465 etc: use of 'conform' instead of 'conforming with'?

L480: Begin new paragraph before 'Site U1387…'

L532: MIS 21e is not between MIS 25 and 22

refs

Hines, S. K., C. D. Charles, A. Starr, S. L. Goldstein, S. R. Hemming, I. R. Hall, N. Lathika, M. Passacantando, and L. Bolge (2024), Revisiting the mid-Pleistocene transition ocean circulation crisis, *Science*, *386*(6722), 681-686.

Wharton, J. H., M. Renoult, G. Gebbie, L. D. Keigwin, T. M. Marchitto, M. A. Maslin, D. W. Oppo, and D. J. Thornalley (2024), Deeper and stronger North Atlantic Gyre during the Last Glacial Maximum, *Nature*, *632*(8023), 95-100.

---

## Author Response (AR1)

Dear editor Erin McClymont,

We are resubmitting our manuscript on the EMPT surface water records at IODP Site U1387 and thank you for considering it for potential publication in Climate of the Past.

Besides the changes pointed out below in the point-by-point responses to the comments received and the respective actions taken, we also updated the data availability statement to link to the data files now available in the world data center PANGAEA. Data access to everyone will be provided as soon as the manuscript is published in Climate of the Past.

Initial author responses during the discussion are given in blue and actions taken in the revised version or additional author responses in **bold and blue**. Please note that in the tracked changes manuscript the Word program is jumping over some lines in the continuous numbering and that line numbers listed in rebuttal comments refer to those we see when opening the document in Word.

Dear authors,

Thank you for your replies to the two reviewers. Although Reviewer 1 was very positive and gave a few details for improvement, Reviewer 2 gave detailed and constructive comments regarding the interpretation of the %GTS proxy as well as some of your arguments for how to interpret the wider results in terms of the subtropical gyre strength/position. From many of your replies there are ways that you can incorporate this feedback into a revised manuscript.

Can you take care to ensure the following in a revised manuscript:
- the details and caveats of the recently developed %GTS proxy are clear. This may require a few lines of text beyond just a citation of the map which you propose to include in the Supplement. In response to L140 (reviewer 2) please also check whether you are emphasising %GTS or whether it is the multi-proxy assessment of the site which is driving your interpretations;
**See our responses to reviewer 2 and actions taken regarding the % GTS proxy below.**

- that there is a more nuanced presentation of d13C as an indicator for AMOC, perhaps drawing further on the details outlined in Muglia and Schmittner (2021) who consider the potential for strength and depth to be recorded, and flag potential location-specific patterns?
**We have added the main outcome of the Muglia and Schmittner study in the introduction (lines 92-95 and 130-131 in the tracked changes manuscript) and added more details on the North Atlantic sites to which we are referring in the cited references in the discussion (lines 781-782, 879-880, 915-917 in tracked changes manuscript). In the initial manuscript we had already used AMOC depth instead of AMOC strength, although without citing the Muglia and Schmittner reference. Since we are using benthic $\delta^{13}$C just as supporting evidence for North Atlantic conditions during the (terminal) stadial events and are not showing (discussing) the benthic $\delta^{13}$C record for intermediate water depth Site U1387, we are declining to go further into a discussion on benthic $\delta^{13}$C signals in the deep (North) Atlantic. We perceive that to be an important discussion in view of the new Hines et al. (2024) results, but it is out of the scope of our manuscript, which focuses on surface water evidence.**

Finally a question about the alkenones: you have some very cold SSTs using UK'37. Have you tried to detect and quantify the C37:4 alkenone? Papers from the Iberian Margin by Belen Martrat for the late Pleistocene showed some interesting relationships between this alkenone and meltwater/IRD events further north and west. You mention that these signals have been identified at a site to the south (U1385) but did you detect this here too? Your graphic comparing your site and U1385 SSTs do not always align: could this be due to variable C37:4 inputs?

**We have C37:4 alkenone data for Site U1387. In general, C37:4 alkenones appear during the terminal stadial events of MIS 26 (T XII) and MIS 22 (TX) and during the early MIS 22 cold event and some of the stadials during the MIS 25 to MIS 24 transition. In many intervals there is high variability in the % C37:4 values from one sample to another. So, we do not have much confidence in the reliability of the data. That is also the reason why Bajo et al. (2020) did not use C37:4 data from Site U1387 and just relied on the Site U1385 record to infer freshwater input on the Portuguese margin (Rodrigues et al., 2017).**

**To reconstruct the UK'37-SST at Site U1387, we applied the UK'37 index, excluding tetraunsaturated alkenones, following the same methodology used at Site U1385 (Rodrigues et al., 2017). Both SST estimations are therefore independent of the concentration of tetraunsaturated alkenones, even if the cold and freshening events are characterized by an increased percentage of C37:4 alkenones.**

**The major misalignment is during the terminal stadial event of MIS 24, which is potentially an age model problem. The MIS 24-MIS 23 interval was poorly constrained in the initial age model by Hodell et al. (2015), which was used in Rodrigues et al. (2017). So, ages in that particular interval might have changed from Hodell et al. (2015) to Hodell et al. (2023), the age model that is now used for the Site U1387 record.**

Additional private note (visible to authors and reviewers only):
Dear authors - thank you for responding to the comments from the reviewers. Please ensure that all of the comments are incorporated into the manuscript: the feedback has been very constructive so I encourage you to draw on those suggestions.
Thank you, and best wishes. Erin.

Comments from reviewer #1
1-Shorten the title: **the title has been changed to**
**The Early – Middle Pleistocene Transition in the Gulf of Cadiz (NE Atlantic) – an interplay between subtropical gyre and extremely cold surface waters**

2-First sentence of the abstract: Clarify that this is change in the dominant cyclicity exhibited by proxy records - **DONE**

3-Line 22: should it read: planktic foraminiferal ecosystems? **Changed and text adjusted accordingly.**

Comments from reviewer #2

More specifically, the authors make use of the trunc coiling direction proxy to support their arguments yet this approach appears quite speculative and gives inconsistent results (e.g. MIS 1 versus MIS 11c according to Billups et al., 2020). Furthermore the proxy has not been demonstrated in the region under study here and it is not at all clear that the %GTS record presented here can be interpreted in terms of gyre strength. I note that the absolute counts of trunc are generally very low (Fig. 4C, typically <10). Kaiser et al., 2019 suggested that low counts should not be relied on and considered only those samples with >20 trunc counts.

The %GTS proxy is a new one and our study is the first to apply it outside of the group of Katharina Billups. The proxy has not been demonstrated for the eastern boundary region of the North Atlantic subtropical gyre but when defining the proxy Billups et al. (2016) took evidence from the complete North Atlantic into account (see their figure 2). The patterns observed by Billups et al. (2016) is confirmed by the updated map below (Figure 1), which is using the % *G. truncatulinoides* right and % *G. truncatulinoides* left data from the Salgueiro et al. (2014) modern analog database to calculate %GTS (keeping in mind that total *G. truncatulinoides* count numbers in the surface samples might not fulfill the cut-off criteria defined by Kaiser et al. (2019); we will include this map in the supplementary material of our revised manuscript as it provides evidence for the eastern boundary current region). The oceanographic conditions in our study region are strongly linked to the eastern boundary circulation of the North Atlantic subtropical gyre and has, in addition, through the Azores Current a direct link to the Gulf Stream. The Azores Current influence on the southern and southwestern Portuguese margin is clearly reflected in Figure 1 by the generally mid-range %GTS values. So, we are confident that the proxy can be applied in our study region.

From the planktonic foraminifera species composition on the Portuguese margin we know that subtropical gyre circulation was not exactly the same during MIS 1 and MIS 11c, because deep dwelling tropical species *G. menardii* and *S. dehiscens* are present in samples from MIS 11c (and MIS 9e; e.g., Voelker et al. 2010) but are not found in MIS 5e or MIS 1 samples (A. Mega, unpublished data). So, we would argue that the differences observed between MIS 1 and MIS 11c are not inconsistencies but changes in the (subsurface?) subtropical gyre that we actually have to explore further (and based on a wider site distribution) in the future.

We thank the reviewer for the comment on the absolute counts! After verification (also with the data uploaded to PANGAEA) it became obvious that Figure 4C was not showing the total counts of *G. truncatulinoides*, but the numbers of *G. truncatulinoides* left only! This was now been corrected and is shown in the updated version of Figure 4 below (as Figure 2).

Since the count numbers are low during the %GTS minimum during the overall peak at the MIS 22/MIS 21 transition we need to revise the text in our discussion and exclude this part. Also, the peak at 918 ka will have to be excluded. We intend to update Figure 7 and show there for U1387 just the reliable %GTS peaks, making it easier for the reader to compare the records, because with the exclusion of that central section of low values, the overall pattern at Site U1387 becomes much more similar to the data of Sites 1058 and 607. We thank the reviewer for reminding us to keep the count numbers in mind, which we had overlooked in that part!

The record of %GTS presented here also shows an inconsistent link with glacial/interglacial climate, with high values observed during both warm and cold periods (although more dominantly during cold periods). And high numbers of shells are only observed when GTS is

high, which could mean that the thermocline is deep (gyre is healthy?) only during those times.

As pointed out by the reviewer many of the %GTS maxima coincide with stadial phases of the millennial-scale climate variability and the gyre strengthening is probably related to the contraction of the gyre. Only the start of MIS 21g and the first peak during MIS 28b, which is, however, less reliable because the two higher % points are based on count numbers at the cut-off limit or just below it, coincide with warmer climate phases. Higher numbers of *G. truncatulinoides* are not only associated with high % GTS as now evident from the correct data plotted in Figure 2 (revised Figure 4). In the manuscript figure panel 4C was actually showing the number of GTS; so, based on that the reviewer's statement is correct.

Yet the authors seem to suggest that the dominance of both right and left coiling varieties can be taken as evidence for a strong /healthy subtropical gyre. For example, (L570) "the presence of *G. truncatulinoides,* especially in its right coiling form, supports subtropical gyre influence during much of the studied interval" and later, (L613) "When the subarctic front receded northward after the terminal stadial event of Termination X, the subtropical gyre expanded again as evidenced by the % GTS maxima at Site 607 and U1387 (Fig. 7), facilitating subtropical water transport to the north and deep-water convection in the North Atlantic (Fig. 8G) (Hodell and Channell, 2016; Hodell et al., 2023a; Kaiser et al., 2019) and the establishment of interglacial conditions." I think I am not mistaken in assuming that some sort of the subtropical gyre will always exist but with variations in position and intensity. Yet I cannot see how the results presented here help with understanding those variations and the authors' discussion offers little illumination. I am sorry that my words seem harsh but I have read the Billups and Kaiser papers and read through the text more than once and still am left wondering what I have learned from the record %GTS.

The statement in Line 570 is related to *G. truncatulinoides* as a subtropical species in general. As shown in Figure 1 the study region has both left and right coiling specimens in the surface sediments. We will delete the "especially in its right coiling form" phrase in the revised manuscript to avoid confusion for the reader. We agree with the reviewer that some form of subtropical gyre will always have existed. The %GTS data offers additional evidence on North Atlantic subtropical gyre conditions and we don't want to suppress that evidence, even if parts of the scientific community like the reviewer might not (yet) see its validity. We hope with including Figure 1 in the supplementary material and with revising the discussion text for the Termination X interval that the evidence and text becomes clearer.

**Actions taken in the revised manuscript:**
1) **Map with the larger % GTS database for the surface sediments added in the supplementary material and related text and more details on the proxy itself added in the method section in lines 283-289 and 295-296 of the tracked changes manuscript.**
2) **Potential inconsistencies with other reconstructions (e.g., Wharton et al., 2024) acknowledged (lines 174-176, 741-742). We also toned the wording down from "strength" to "conditions" within the subtropical gyre in the introduction and later on in the % GTS discussion chapter and in the conclusions (lines 1071-1074).**
3) **Results section on the % GTS data fully rewritten (and condensed), which now also mentions the ≥20 specimens cut-off criteria defined by Kaiser et al. (2019)**

and how it affects our data. Criteria also mention in the caption for figure 4 and methods.

4) **Figures 4 has been corrected to now show the true number of specimens counted to calculated the % GTS data.**
5) **Figure 7 has been adjusted to show only the reliable % GTS data points for Site U1387.**
6) **The discussion of the % GTS evidence has been modified**
   a) **for clarity regarding the geographical distribution of the dextral coiling variant (line 660-661);**
   b) **cleaned up for confusing/irrelevant text (e.g., paragraph starting in line 666)**
   c) **incorporated other faunal evidence, e.g. the AzC factor fauna**
   d) **new text with explanation for the early, less pronounced % GTS peaks**
   e) **corrected text for the reliable data at Site U1387 for the MIS 22a % GTS peak**
   f) **new text for the MIS 24 and MIS 21 maxima.**

Some of the results section is overly descriptive and potentially verbose. Consider condensing to save space and improve readability.

We will condense the results section in the revised version.

**We condensed the section on the % GTS data and made a small adjustment in lines 471-472. We did not make further cuts, because the sections contain information we perceive important, e.g., all the information on the planktonic foraminifera fauna will be relevant for ecosystem studies that might use our data, and mentioning of error bars for potential comparison with climate model outcomes.**

Line 40/41: What is the evidence for this sequence of events? You imply that extreme AMOC weakening caused the migration of the subarctic front but could it not be the other way around?

There is no concrete evidence per se as the age model errors from the various IODP Sites in the North Atlantic do not allow to deduce a true sequence of events. Ice-rafting event and evidence for a shallower AMOC, i.e., lower benthic $\delta^{13}$C values, are contemporary at IODP Site U1308. However, the statement in the abstract is written from the perspective of the southern Portuguese margin, meaning the subarctic front was displaced (assuming an interglacial position similar to today) from a latitude of 51°N (or even further north in the eastern basin) to a latitude of 37°N. In our opinion, such an extreme displacement is not possible just by (initial) freshening in the subpolar gyre and the associated southeastward displacement of the subarctic front. One needs the "amplification" of a reduced overturning circulation and associated reduced heat and salt flux to the more northern latitudes (i.e., a weakened and/or "displaced" North Atlantic Current) to push the subarctic front as far south as the latitudinal range of the southern Portuguese margin.

**We have adjusted the sentence to be clearer that cooling and freshening in the higher latitudes also played a role.**

L53: We don't yet know what caused the transition to more intense and longer-lasting glacials although $CO_2$ and ocean circulation changes were intimately involved. Perhaps tone down the implied certainty of cause and effect here.

**Toned down; "ultimately resulted in" replaced by "led to".**

L56: A recent paper by Hines et al. [*Hines et al.*, 2024] suggests that no substantial change in glacial deep ocean circulation occurred across the MPT. This requires us to rethink some of our assumptions about ocean dynamical changes across the MPT and should be cited (and discussed – see later).

We will add Hines et al. (2024) as additional reference and check if there are instances where their suggestions might be included in the discussion later on.

**We modified the text starting in line 64 in the tracked changes version and added a sentence summarizing the new findings of Hines et al. (2024). We also refer to their findings in lines 98-100 and 103-105 (tracked changes version). The wording was also toned down in the abstract (line 17).**

L93: Again, there is emphasis placed on changes CAUSED by AMOC variations across the MPT but this order of cause and effect needs to be reconsidered.

**We deleted the link to the AMOC as it is not essential in that part of text.**

L140: The study relies heavily on the use of trunc coiling direction as a proxy for gyre circulation as described by Billups et al., 2016 (which I think implies a weaker gyre circulation during the LGM). A recent study by Wharton et al., [*Wharton et al.*, 2024], using sites in a similar region as the Billups study, suggests that the glacial gyre was stronger and deep than today. This paper needs to be considered in the light of how reliable the trunc proxy can be.

The %GTS proxy is just a small part of all the new data generated for the current study and it therefore does not rely "heavily" on it. However, since it is a new proxy, we will add the cautionary note that there might exist inconsistencies with other reconstructions (e.g., Wharton et al., 2024) that need to be explored further in the future (which is outside of the scope of the current manuscript).

**We are now referring to the inconsistencies between the two types of gyre reconstructions in lines 174-176 and 741-742 of the tracked changes manuscript and toned down the wording related to the Kaiser et al. (2019) results in the conclusions (line 946).**

L218: Please give details of how the age model used here differs from the Bajo study. The Bajo study was defined by it's age model (tied to U1385 and speleothem records) and changing it here would seem to contradict the Bajo study.

This comment does not make much sense for our manuscript because the Bajo et al. study and thus age model does not cover the complete interval presented here. The tuning to the speleothem chronology also led to some extreme changes in the sedimentation rates. We will add a figure in the supplementary material to show the age differences between the Bajo et al. speleothem ages and the LR04/Probstack ages. We have never compared the two age model

approaches (U/Th dated vs. orbitally tuned), but do not expect major differences. Similar to the Site U1385 record, our tuning target, the Site U1387 timeseries is much longer than presented here (MIS 17 to MIS 52 for the multi-proxy data) and we therefore established a new age model that allows placing the complete timeseries on a consistent LR04/Probstack chronology.

**The age differences between Bajo et al. (2020) and the LR04 and Probstack chronologies for the U1387 timeseries used here are actually larger than expected and affect the interval between MIS 20 and late MIS 25. So, we added supplementary figure 4 highlighting the differences and added a sentence in subchapter 5.1 (lines 363-366 in tracked changes manuscript) pointing out the existence of those differences. We are, however, not discussing the discrepancies further as that is a topic beyond the scope of this manuscript.**

L241: "the amount of sinistral coiling direction of this species increases when the subtropical gyre circulation is more intense." Specifically, a higher concentration of left coiling truncs supposedly reflects a deeper permanent thermocline and hence healthy gyre circulation. But this depends very much on the site selected. Are the authors confident that their site will work in the same way?

When defining the parameter Billups et al. (2016) used data from the whole North Atlantic - see their figure 2. Our site is located on the eastern edge of the subtropical gyre and is strongly influenced by the Azores Current and the Portugal-Canary Currents which represent the eastern boundary re-circulation of the subtropical gyre. The boundary distinguishing between left and right coiling dominance in the Northeast Atlantic in figure 2 of Billups et al. (2016) actually follows the northern boundary of the Azores Current and if confirmed in the updated %GTS map shown in Figure 1. So, yes, we have confidence that the proxy should work in our region and it was tested here for the first time. The coincidence of %GTS peaks at Site 607, 1058 and U1387 (our figure 7) seems to confirm this (initial) assumption.

**We added the map (and some text) showing the expanded %GTS data based on the Salgueiro et al. (2014) surface sediment database in the supplementary material and added more information on the proxy in the methods as support of our confidence in applying that proxy in our study region.**

L465: I wouldn't disagree with cooling bit but I don't think we know what the climate sensitivity would be.

We will adjust the wording here to make clear that we mean this statement in a general sense of "a cooling trend" to be expected and not that climate sensitivity during the LGM was the same as during the MPT glacials.

**Yes, our use of "similar" was misplaced and we have adjusted the wording in the revised version (lines 553-554 in tracked version manuscript).**

L476: so are the waters at the site actually warmer or is it just an illusion? And if so, then should you plot the faunal SST at all?

Based on the faunal evidence the waters were warmer than, for example, many of the late Pleistocene glacial periods. The higher percentage of subtropical species is creating the

"bias", but also indicates that the waters were relative warm and contained a relevant contribution of subtropical gyre waters. The questions are just how warm or how gradual the cooling was at the zooplankton level; questions which might only be solved by Mg/Ca or clumped isotope temperature reconstructions as the lipid biomarker SST might include their own caveat (e.g., Ausin et al., 2021), which we did not discuss in the manuscript as the focus is on the foraminifera faunal data. In our opinion, it makes sense to show the faunal SST -not only to highlight the probable bias, which also affects other studies in the region (e.g., Martin-Garcia et al., 2015), but also because some "truth" is hidden in the data, which should not be ignored.

Ausín, B., Magill, C., Haghipour, N., Fernández, Á., Wacker, L., Hodell, D., Baumann, K.-H., Eglinton, T.I., 2019. (In)coherent multiproxy signals in marine sediments: Implications for high-resolution paleoclimate reconstruction. Earth and Planetary Science Letters 515, 38-46, https://doi.org/10.1016/j.epsl.2019.03.003.

**We have not changed the text or added a comment on why we are keeping the SST-PF in the manuscript because the arguments presented above are generally addressed in the text of lines 556-572 in the tracked changes manuscript. We added, however, the clarification that the faunal evidence itself is indicating cooling (line 561).**

L492: But Uk37 temp was as high during MIS 27 as e.g. MIS 21.

We will adjust the wording here to make clear that the cooler MIS 27 statement refers to the fauna derived SST estimates.

**We added the clarification that the cooler SST in MIS 27 statement is only valid for the PF-SST (lines 587-588 in tracked changes manuscript).**

L507: earlier (L458) you stated that interglacials were associated with stable SSTs

We thank the reviewer for pointing out this inconsistency in our text. We will adjust the wording in line 458. **DONE (lines 547-548 in tracked changes manuscript)**

L512: the difference between faunal and Alk SST is important and should be further discussed (is it to do with the gyre again?). Also the comparison with MIS 11 is interesting but needs to be filled out a lot.

We will add a bit more text here but will not go for a full comparison with MIS 11, which is beyond the scope of the current manuscript. The early phases of MIS 25e and MIS 11c were both associated with the evidence for (more persistent) upwelling. So, the difference is more caused by ecological preferences of the studied phyto- (coccolithophores) and zooplankton (foraminifera) groups than the gyre circulation.

**We added a sentence stating that both MIS 25e and MIS 11c experienced upwelling activity (lines 610-613), which likely caused the cooler SST.**

L515-519: this is interesting, since MIS 28b was only missed at more northerly sites and was warm at the southern margin of the Atlantic Inflow and at the site described here. Could the authors comment on the conclusions of Barker et al. (2021) on how surface ocean circulation might have changed across the MPT?

Since it is out of the scope of the current manuscript, we will not comment on the conclusions of Barker et al. (2021) in it. From Site U1387 we have now faunal evidence, in particular related to *N. pachyderma*, that MIS 29/MIS 28 (prior to the data presented in the current manuscript) might have been a critical period in species variant evolution in the mid-latitudinal N Atlantic, in agreement with the general 1.1-1 Ma time frame of Huber et al. (2000) for the emergence of the polar variant of *N. pachyderma.* As the Barker et al. (2021) study relies soly on % *N. pachyderma* evidence one might have to be more careful in the data interpretation, especially at sites under subtropical gyre (607) and North Atlantic Current (U1304, ODP 980/981) influence. It would probably also be relevant to look into the full planktonic foraminifera fauna and not just the % *N. pachyderma* to obtain a more complete picture of circulation changes/inflow strengthening. We might get back to this topic when the manuscript on the Neogloboquadrinids evolution at Site U1387 will be written (hopefully later this year).

**Nothing added/changed in the manuscript.**

L688: Benthic $d^{13}C$ cannot be interpreted as a quantitative proxy for the AMOC.

The modeling work of Muglia and Schmittner (2021) shows that benthic $\delta^{13}C$ is reflecting more overturning depth and not strength. That is the reasoning we are following with our wording here. We will add the Muglia and Schmittner reference in the revised manuscript to make this clearer.

Muglia, J., Schmittner, A., 2021. Carbon isotope constraints on glacial Atlantic meridional overturning: Strength vs depth. Quaternary Science Reviews 257, 106844, doi: https://doi.org/10.1016/j.quascirev.2021.106844.

**See our response to the editor's comment above.**

L711: this was speculation in the Elderfield paper

We will adjust the wording to point out that this was a speculation.

**We have toned down the wording to "could lead to" (line 903 in the tracked changes manuscript).**

Technical:
L19: Cooler SSTs in general or during glacials? **Yes, during glacial; text adjusted.**
L110: Never proven – replace with demonstrated? **DONE**
L202 etc: Use of 'weighted' instead of 'weighed'- **DONE**
L322: "In total, 16 species were identified (Table 1; Fig. 3), with the diversity of the subtropical fauna appears to be diminished" – please check sense (or lack of) – **Corrected.**
Para beginning L352: This is very descriptive and wordy. Please condense. **This whole results section has been rewritten and the information condensed.**
L389: do you mean extremely short or extremely cold? – **extremely cold; so, corrected to short, extremely cold…**
L465 etc: use of 'conform' instead of 'conforming with'? **changed to "conforming with" in two instances and to "in accordance with" in the other two instances (see tracked changes version of the manuscript)**

L480: Begin new paragraph before 'Site U1387…' **DONE**
L532: MIS 21e is not between MIS 25 and 22 – **Corrected.**

---

## Author Response (AR2)

Dear Editor Dr. McClymont,

We would like to sincerely thank you for accepting our manuscript for publication at the Climate of the Past journal. We appreciate the constructive feedback provided during the review process, which has helped us improve the quality of our work.

We have carefully considered the last set of comments and have incorporated the necessary changes into the manuscript accordingly. All requested modifications have been implemented to ensure alignment with the reviewers' and editors' suggestions.

Please find attached the final revised version of our manuscript. Should you require any further information or additional adjustments, please do not hesitate to contact us.

Thank you once again for your time and consideration. We look forward to the publication of our study at Climate of the Past.

Best regards,

Aline Mega and Co-autors